# Effects of Pressure, Hypoxia, and Hyperoxia on Neutrophil Granulocytes

**DOI:** 10.3390/biom14101242

**Published:** 2024-09-30

**Authors:** Richard F. Kraus, Daniel Panter, Michael A. Gruber, Stephanie Arndt, Petra Unger, Michael T. Pawlik, Diane Bitzinger

**Affiliations:** 1Department of Anaesthesiology, University Medical Centre Regensburg, Franz-Josef-Strauss-Allee 11, 93053 Regensburg, Germany; 2Department of Dermatology, University Medical Centre Regensburg, Franz-Josef-Strauss-Allee 11, 93053 Regensburg, Germany; 3Center of Dive and Hyperbaric Medicine, Department of Anaesthesiology, Caritas Hospital St. Josef, Landshuter Str. 65, 93053 Regensburg, Germany

**Keywords:** neutrophil, hypoxia, hyperoxia, pressure, oxygen, diving, hyperbaric

## Abstract

**Background:** The application of normo- and hyperbaric O_2_ is a common therapy option in various disease patterns. Thereby, the applied O_2_ affects the whole body, including the blood and its components. This study investigates influences of pressure and oxygen fraction on human blood plasma, nutrient media, and the functions of neutrophil granulocytes (PMNs). **Methods:** Neutrophil migration, reactive oxygen species (ROS) production, and NETosis were examined by live cell imaging. The treatment of various matrices (Roswell Park Memorial Institute 1640 medium, Dulbecco’s Modified Eagle’s Medium, H_2_O, human plasma, and isolated PMNs) with hyperbaric oxygen (HBO) was performed. In addition, the expression of different neutrophil surface epitopes (CD11b, CD62L, CD66b) and the oxidative burst were investigated by flow cytometry (FACS). The application of cold atmospheric plasma (CAP) to normoxic and normobaric culture media served as a positive control. Soluble reaction products such as H_2_O_2_, reactive nitrogen species (RNS: NO_2_^−^ and NO_3_^−^), and ROS-dependent dihydrorhodamine oxidation were quantified by fluoro- and colorimetric assay kits. **Results:** Under normobaric normoxia, PMNs migrate slower and shorter in comparison with normobaric hyper- or hypoxic conditions and hyperbaric hyperoxia. The pressure component has less effect on the migration behavior of PMNs than the O_2_ concentration. Individual PMN cells produce prolonged ROS at normoxic conditions. PMNs showed increased expression of CD11b in normobaric normoxia, lower expression of CD62L in normobaric normoxia, and lower expression of CD66b after HBO and CAP treatment. Treatment with CAP increased the amount of ROS and RNS in common culture media. **Conclusions:** Hyperbaric and normobaric O_2_ influences neutrophil functionality and surface epitopes in a measurable way, which may have an impact on disorders with neutrophil involvement. In the context of hyperbaric experiments, especially high amounts of H_2_O_2_ in RPMI after hyperbaric oxygen should be taken into account. Therefore, our data support a critical indication for the use of normobaric and hyperbaric oxygen and conscientious and careful handling of oxygen in everyday clinical practice.

## 1. Introduction

One of the main functions of human blood is to transport oxygen from the lungs to the organs and peripheral tissues [1]. However, the oxygen supply to which people are exposed can vary extremely, both in pressure and oxygen concentration. On the one hand, there are conditions of environmental hypobaric hypoxia, which, e.g., affect climbers breathing ambient air at the summit of Mount Everest [2]. On the other hand, there are conditions of hyperbaric hyperoxia, which people experience during scuba diving into the depths of the sea [3]. Apart from that, there are conditions in which the uptake of O_2_ into the blood, the O_2_ transport within the blood, or the O_2_ release into peripheral tissues are disturbed. Examples are the event of a gas embolism, decompression illness after a scuba diving accident, or an intoxication with carbon monoxide [4,5,6].

In such cases of severe oxygenation disorders, when the inhalation of oxygen-enriched air or even mechanical ventilation with 100% O_2_ may not be sufficient for adequate oxygenation of the blood, hyperbaric oxygen therapy (HBO treatment) might be necessary [7,8]. If sufficient oxygenation cannot be achieved, even through maximal ventilation with HBO, and cardiovascular failure occurs alongside or additionally, (partial) replacement of heart and lung functions by Extracorporeal Membrane Oxygenation (ECMO) can provide additional benefits under certain circumstances [1,9,10].

In all described situations, different O_2_ concentrations and pressures affect the whole blood, not only erythrocytes, as O_2_ carrier cells, but also neutrophil granulocytes (PMNs), as the most mobile and abundant cellular component of the innate immune system [11].

Even in physiologic conditions, oxygen levels drop from the environmental level of 20% to tissue concentrations of about 3–4% (or less). Starting from atmospheric air to individual cells, pO_2_ decreases from approximately 150 mmHg in the upper airway to about 30 mmHg in most tissues and finally to as low as 5 mmHg in peripheral tissues, where these values are quite well maintained. PMNs circulate in the blood for most of their life circle, so these cells are confronted every minute with a maximum of 120 mmHg pO_2_ (with an inspired O_2_ fraction of 0.21) in the arteries and a minimum of 5 mmHg pO_2_ in the smallest venules in the tissue [12,13]. At the sites of inflammation, PMNs leave the blood vessels through the endothelium, a process known as extravasation. This process is characterized by a multitude of changes in the neutrophil surfaces (the most important players are CD11b, CD62L, and CD66b), which ultimately allows the PMNs to adhere to and migrate through the vessel wall. After extravasation into inflammatory human tissue, PMNs migrate along chemotactic gradients in the interstitium and perform specific neutrophil immune functions as a first line of defense of the innate immune system. Once at the site of action, PMNs use the following two fundamentally different mechanisms in addition to phagocytosis of pathogens to defend against infectious agents: oxygen-dependent and independent mechanisms [11].

The most important oxygen-independent antimicrobial mechanisms are the release of lytic enzymes (such as myeloperoxidase) and bactericidal peptides and the release of DNA in the form of reticular structures into the extracellular space (NETosis) [11]. The most important oxygen-dependent mechanism is the formation of reactive oxygen species (ROS) [14]. As part of the so-called “respiratory burst reaction”, phagocytizing PMNs show a strong increase in their oxygen consumption, which is caused by the NADPH-dependent production of superoxide anions (O_2_^−^). Superoxide anions are the trigger that leads to the formation of ROS, i.e., hydrogen peroxide (H_2_O_2_), hydroxyl radical (OH•), and hypochlorous acid (HOCl). These contribute to the destruction of bacteria [15,16].

In this way, as the main part of the respiratory burst (in the following, the terms neutrophil “ROS production” and “oxidative burst” are used as synonyms for the same phenomenon just described), oxygen is involved in one of the most important defense systems of human neutrophils [17]. Thereby, it seems likely that PMNs are influenced by this wide range of different oxygen conditions.

The medical use of oxygen has been viewed increasingly critically over the last decade. Under certain circumstances (such as in acute cardiovascular disease, brain ischemia due to, for example, stroke, shock, or carbon monoxide intoxication), the positive effects of oxygen application may be claimed simply by relief of assumed local tissue hypoxia [18]. However, excessive oxygen supplementation may have detrimental pulmonary and systemic effects because of enhanced oxidative stress and inflammation [19]. The role of neutrophils in this process has already been investigated in a few studies, leading to ambiguous results [20,21,22,23]. Therefore, we conducted this study to further elucidate the influence of different hyperbaric pressure, hyperoxia, and hypoxia on neutrophil function. Thereby, different aspects of neutrophil function were considered. Figure 1 gives an overview over the used methods and investigated parameters.

We performed live cell imaging (LCI) chemotactic experiments and determined neutrophil migration, ROS production, and NETosis. Thereby, two different experimental setups were used. Simultaneous live cell imaging was performed to measure the influence of different oxygen concentrations over a long period, whereas LCI after pressure treatment was performed to measure the effect of short-term oxygen influence. Moreover, changes in surface epitopes were examined to measure of neutrophil activation.

By activation, PMNs can improve their chemotactic migration ability to reach affected areas more quickly while increasing their ROS production, which leads to an enhanced antimicrobial effect. After maximum ROS production, migration ends and is replaced by NETosis. In the case of increased activation, the subsequent processes are expected to be faster. Furthermore, neutrophil activation can lead to changes in surface epitopes, which was investigated by FACS analysis. Thereby, the expression of CD11b, CD62L, and CD66b on PMNs is typically regarded as an indicator of the cells’ activation statuses. While CD11b and CD66b are upregulated upon PMN activation, CD62L levels decrease in response to neutrophil stimulation [24]. In addition, the influence of different O_2_ conditions on nutrient media (RPMI, MEM, H_2_O, blood plasma) was investigated by ROS and RNS measurements.

## 2. Results

### 2.1. Normobaric Simultaneous Live Cell Imaging

#### 2.1.1. Analysis of Neutrophil Migration

To investigate the long-term influence of normobaric oxygen on different neutrophil functions including chemotactic migration, ROS production, and NETosis, simultaneous LCI was performed. Thereby, a total of *n* = 10 experiments (based on 10 individual test persons) with the three-chamber IBIDI-Slides of simultaneous live cell imaging (that means 20 individual blood samples) and a total of *n* = 50,368 PMN cell migration tracks were included in the results. Thereby, in 4% and 50% O_2_, nine IBIDI channels each were included in the analysis. In 83% O_2_, there were 12 successfully evaluated channels. All channels included in the evaluation were successfully compared simultaneously with channels of 20% O_2_.

In normobaric hypoxia (4%) and hyperoxia (83%), migration was higher than in the other oxygen fractions (see Figure 2). This observation was most evident in the first time slot because statistically significant differences were found in all pairwise combinations of the TL parameter with the exception of the combination of 50% and 20% oxygen, as well as the combination of 4% and 83% (*p* < 0.001, see Table 1). The results of the other time slots were similar to the results of the first time slot and showed comparable significant findings. A detailed statistical evaluation of the individual subgroups as well as the evaluation of the TrackSpeed parameter is described in Appendix D (see Table A3 and Table A4). In all O_2_ concentrations, the determined TrackLength diminished with the increasing duration of the experiment. The total number of single moving neutrophil cells (tracks) decreased with increasing observation time from slot to slot (see Table 1).

To summarize, hypo- and hyperoxic conditions enhanced neutrophil chemotaxis immediately after fMLP stimulation, with PMNs migrating the farthest and fastest in hypoxia (4% O_2_). Chemotaxis was initially largest and equalized in later observation periods (time slots).

#### 2.1.2. Analysis of Neutrophil ROS Production

In normobaric hypoxia (4%), T_max_ROS was determined at a mean value (MV) of 96.5 min ± 36.1 min [*n* = 8]. T_max_ROS showed no significant differences (*p* > 0.05) to normobaric normoxia 20% O_2_ (MV 98.7 min ± 33.0 min [*n* = 24]) or normobaric hyperoxia 50% O_2_ (MV 101 min ± 38.3 min [*n* = 8], 83% O_2_ MV 99.0 min ± 31.0 min [*n* = 11]; see Figure 3).

#### 2.1.3. Intensity of Neutrophil ROS Production (ROS Intensity)

The ROS intensities [RFI] plotted against the 10 min interval times qualitatively showed a uniform curve progression with an initial increased intensity over approximately 60 min that decreased as the observation progressed. After three hours, the intensities appeared at a constant low level (see Figure 4). The lowest initial RFI with a slight decrease in intensity over the course of the observation period was observed at the 4% oxygen concentration. The intensities at the 20% oxygen concentration were at a medium level, with a moderate decrease in intensities over time. The highest RFI at the beginning with a strong reduction over time was observed at the 50% oxygen concentration. The initial RFI at the 83% oxygen concentration was a little above that at the 4% oxygen concentration, and a moderate decrease in RFI was observed.

A quantitative examination (see Figure 5) at time points t = 10 min and t = 30 min revealed significantly different results for the 4% vs. 20%, 20% vs. 83%, and 50% vs. 83% concentration groups (see Figure 5). The results for oxygen concentrations of 4% vs. 83%, 4% vs. 50%, and 20% vs. 50% did not reach statistical significance (*p* > 0.05). RFI was highest in the 20% oxygen concentration group (RFI = 52.0 (t = 10 min); RFI = 49.0 (t = 30 min)). At time point t = 60 min, the RFI of the 20% vs. 83% oxygen concentration differed significantly. At this time point, no other statistically significant differences were observed (*p* > 0.05). Nevertheless, at t = 60 min, the highest intensities (RFI = 44.0) were also observed in the 20% oxygen concentration group.

In summary, in all groups, higher intensities were initially measured, which decreased with time. In normobaric normoxia (20% O_2_), this decrease reached a higher plateau after about 140 min, while in the other groups, it kept falling until about 200 min.

### 2.2. Analysis after Pressure Treatment

To investigate the short-term influence of normo- and hyperbaric oxygen on different neutrophil functions including chemotactic migration, ROS production, and NETosis, LCI after pressure treatment was performed. For clarity, the terms normoxia, hyperoxia, HBO, and CAP are used to list the results.

#### 2.2.1. Migration Results of Live Cell Imaging after Pressure Treatment

For the TrackLength parameter after pressure treatment, a total of *n* = three experiments (based on three individual test persons) with a total of *n* = 634 PMN cell migration tracks were included in the results. Significantly lower values were achieved in the normoxia group in all time slots than in the other two conditions (*p* < 0.001, see Figure 6). Additionally, in the fourth time slot (181–240), the TrackLength of the hyperoxia group was significantly higher than in the HBO group (*p* = 0.03). Within the groups, TL showed constant values over 3 h and decreased only thereafter (see Table 2). This was reflected in the significant differences that could be found, above all, between the first three observation time slots and the last three observation time slots (see Table A6 in Appendix E). The TrackSpeed in the normoxia group was significantly lower than in the other two groups (*p* < 0.001). Additionally, in the fourth time slot (181–240), TS was higher in hyperoxia than in HBO (*p* = 0.28, for details, see Table A5 in Appendix E).

To summarize, in LCI after pressure treatment, migration was significantly lowest in the normoxia group in all time slots (see Table 2). There were few differences between normobaric hyperoxia and HBO, whereby the fourth period of the LCI after pressure treatment was TL_Hyperoxie_ > TL_HBO_ (see Table 2).

#### 2.2.2. Analysis of Oxidative Burst and NETosis by Live Cell Imaging after Pressure Treatment

No significant differences in the T_max_ROS parameter and ET_50_NETosis could be determined after different pressure treatments (*p* > 0.05).

#### 2.2.3. Analysis of Neutrophil Surfaces and Oxidative Burst by FACS after Pressure Treatment

To investigate the short-term influence of normo- and hyperbaric oxygen on neutrophil surface epitopes as a measure of neutrophil activation, flow cytometric measurements were performed. FACS analysis after pressure treatment resulted in a significantly higher MFI of CD11b (see Figure 7a) in normoxia than in hyperoxia and in the CAP group. The difference between the HBO and CAP groups did not reach significance (*p* = 0.052). The MFI of CD62L (see Figure 7b) showed significantly lower values in normoxia than in hyperoxia and in the CAP group. The MFI of CD66b (see Figure 7c) had significantly lower values in the CAP group than in the HBO group and in the normoxia group. When PMNs were activated by fMLP/TNFα, the lowest MFI (Rhodamine 123) was measured in the normoxia group (significant differences with MFI (Rhodamine 123) in hyperoxia, MFI (Rhodamine 123) in HBO, and CAP). There were no significant differences among the other three groups (*p* > 0.05). With PMA activation, MFI (Rhodamine 123) was significantly higher in hyperoxia than in normoxia. In the CAP group, it was significantly higher than in normoxia. Overall, MFI (Rhodamine 123) was significantly higher in the PMA group than for fMLP/TNFα except in the normoxia group (see Figure 7d).

To summarize, PMNs showed an increase in CD11b in normoxia in comparison with the other oxygen conditions. CD62L showed significantly lower levels in normoxia, and CD66b had lower expression in the CAP group than in the HBO group and in the normoxia group. Oxidative burst was higher in the CAP group than in the hyperoxia and normoxia groups.

#### 2.2.4. Results of H_2_O_2_, Rhodamine 123, NO_2_^−^, and NO_3_^−^ Measurements in Different Matrices

To investigate the short-term influence of different O_2_ conditions on nutrient media in which experiments with PMNs are usually performed (RPMI, MEM, H_2_O, blood plasma), ROS and RNS measurements were performed. For clarity, the results of H_2_O_2_, Rhodamine 123, NO_2_^−^, and NO_3_^−^ measurements are summarized below. A detailed presentation of the statistical results can be found in Appendix F in the supplement.

(a)The highest concentration of H_2_O_2_ in DMEM and RPMI (see Table 3) was measured in the CAP group (all *p* values < 0.033). In H_2_O, plasma, and PMNs in plasma, no significant differences could be determined for H_2_O_2_ (see Figure 8).(b)In DMEM and H_2_O, the level of Rhodamine 123 fluorescence was the highest in the CAP group (all *p* < 0.040). In RPMI, Rhodamine 123 fluorescence was significantly higher in CAP (all *p* < 0.01). In plasma, Rhodamine 123 fluorescence was significantly higher in CAP than in normoxia (*p* = 0.013).(c)In DMEM and H_2_O, NO_2_^−^ was highest in the CAP group (all *p* < 0.041). In RPMI, NO_2_^−^ was highest in the CAP group (*p* < 0.004). In plasma and PMNs in plasma. NO_2_^−^ was highest in the CAP group (all *p* < 0.045).(d)In DMEM and H_2_O, NO_3_^−^ was highest in the CAP group (all *p* < 0.044). In RPMI, NO_3_^−^ was highest in the CAP group (*p* = 0.003).

To summarize, H_2_O_2_ was highest in RPMI, followed by DMEM, whereby the difference in the amount of H_2_O_2_ in RPMI compared with the other media was greater than that triggered by pressure and hyperoxia (see Figure 8). ROS were lowest overall in human plasma. It made no difference whether PMNs were in the plasma or not.

## 3. Discussion

The aim of the present study was to investigate the influence of different pressures and oxygen concentrations on neutrophil function under standardized in vitro conditions. We found significant influences of various oxygen conditions on neutrophil functions, which are summarized below (see Table 4) and placed in the context of the existing literature.

### 3.1. PMN Migration Behavior Dependent on Oxygen Concentration and Pressure

During simultaneous LCI, which was performed exclusively under normobaric conditions, hypo- and hyperoxic conditions appear to cause very strong neutrophil chemotaxis directly after fMLP stimulation (recognizable by the significantly highest migration distances at 83% O_2_ in the first section (TL = 131.8 µm [IQR = 145.3 µm]) and 4% O_2_ (TL = 115.6 µm [IQR = 178.2 µm])) (see Table 1 and Table 2). Here, PMNs migrated the farthest and fastest in hypoxia (4% O_2_), which was evident in this group, especially in the second section with the highest values of TL (97.7 µm [IQR = 147.8 µm]) and TS (0.11 µm/s [IQR = 0.13 µm/s]). The neutrophil chemotaxis of the hypo- and hyperoxic conditions was initially largest and equalized with the other groups at later observation time slots (recognizable by lack of significant differences between 4% and 50% O_2_ in time slots 121–180 and 181–230).

In both simultaneous LCI and LCI after pressure treatment, migration was significantly lowest in the normoxia group in all time slots (see Table 1 and Table 2). There were few significant differences between normobaric hyperoxia and HBO, whereby the fourth period of the LCI after pressure treatment was TL_Hyperoxie_ > TL_HBO_ (see Table 2).

In all our experiments, neutrophil chemotaxis appeared to be greater in hyperoxic conditions than in normoxia, regardless of whether PMNs were exposed to 100% O_2_ for a longer period or only briefly for 5 min during pressure treatment. The pressure component seems to have less effect on PMN migration behavior than the O_2_ concentration in the vicinity of the PMNs.

Since former studies showed that higher oxygen concentrations in particular led to stronger tissue inflammation, the increased neutrophil migration in hyperoxia seen in our experiments could be a sign of increased PMN immune activity in hyperoxic conditions. In this context, Schwartz et al. provided evidence that HBO increases the phagocytic activity of PMNs and enhances the killing of *S. aureus*, which promotes an immune-enhancing component of HBO [25]. However, data is controversial. Notably, Kiers et al. described that short-term hyperoxia has no effect on phagocytosis, and Gadd et al. observed higher oxygen tensions to have no effect on PMN phagocytosis [21,26]. Moreover, Grimberg-Peters et al. reported that HBO did not influence neutrophil chemotaxis or apoptosis (in a transwell assay) [27].

In contrast, the hypoxic effect could be explained by the fact that low oxygen concentrations most closely approximate physiological conditions in peripheral tissue and therefore lead to greater migration. In particular, arterial end-stream regions are locations where—depending on the perfusion situation—physiologically very low oxygen partial pressures usually prevail. For skeletal muscle, these are about 12–70 mmHg (1.6–9.3 kPa; 1.6–9.2%) [28]. Circulating PMNs come into contact with an oxygen partial pressure of about 159 mmHg (21 kPa) in the alveoli in purely physiological terms (20.9% oxygen content of room air at 101 kPa air pressure at sea level), whereas the arterial oxygen partial pressure is usually between 9 and 13 kPa (70 and 100 mmHg) [29]. Moreover, hypoxia is a key feature of chronic wounds, and inflammatory sites are characterized by low levels of oxygen and glucose and high levels of reductive metabolites [30]. The destruction of vasculature in the wound and high oxygen consumption by inflammatory cells, such as neutrophils, result in low oxygen levels [31]. Under hypoxic conditions, neutrophil apoptosis and subsequent clearance are markedly inhibited [30,32]. This apparently helps PMNs survive longer at inflammatory sites [30].

All groups showed a tendency of a decrease in track numbers and TL over the whole observation period, independent of pressure and oxygen, which is in line with previously conducted PMN chemotaxis experiments [33,34,35]. Since PMNs usually underwent NETosis at the end of our experiments, the observation period ended at this point.

### 3.2. ROS Production and NETosis

Variation in O_2_ concentration or pressure (to which PMNs are exposed for longer periods or treated for short periods) had no effect on the time point of maximum ROS production or NETosis in our experiments.

#### 3.2.1. ROS during Simultaneous LCI

Nevertheless, the course of ROS intensity (the measurement of total ROS production detected by the transformation of DHR into Rhodamine 123 in a single cell) differed in simultaneous LCI with increasing duration of the experiments. Initial higher intensities were measured in all groups, which decreased towards the end. In normobaric normoxia (20% O_2_), ROS intensity was constant until t = 160 min, while in the other groups, ROS intensity continued falling (see Figure 5). Thus, the power of ROS production in individual PMN cells seems to be greatest under normobaric and normoxic conditions (recognizable by the highest intensities at 20% and 50% O_2_ at t = 10 min, t = 30 min, and t = 60 min (see Figure 5)). Our data do not exactly match the results of Kiers et al. and Gadd et al., who reported that short-term hyperoxia or higher oxygen tensions have no effect on neutrophil ROS production or oxidative burst [21,26]. Nevertheless, our data do not contradict theirs either. Moreover, it is possible to associate a lack of a decrement in the ROS intensity in normoxia with a prolonged PMN defense response. Nonetheless, a long-term application of normobaric hyper- or hypoxia leads to a faster decrease in ROS production in single neutrophil cells, whereas—according to Gadd et al.—short-term oxygen exposure and intermittently elevated oxygen tensions do not lead to exhaustion of PMNs’ ability to undergo oxidative burst [26].

The “DETO2X-AMI” trial and the Australian “AVOID” trials both concluded that oxygen therapy in acute myocardial infarction can lead to tissue damage through an enhanced respiratory burst [23,36]. In particular, these findings are consistent with our observation that ROS production occurs to an increased extent at intermediate oxygen concentrations. When oxygen is applied by inhalation, inspiratory oxygen concentrations of about 60–70% are achieved with an oxygen mask, which is commonly used in prehospital emergency care [37]. These were enacted between the 50% and 83% oxygen concentrations in our study, where we observed increased ROS production. To sum up, although the time of maximum ROS production for all cells (T_max_ROS) did not change significantly, it appears that the influence of O_2_ concentration is likely at the single cellular level. Under hypoxia and hyperoxia, the ROS production of single neutrophil cells decreases more rapidly under hypoxia and hyperoxia. Moreover, the individual PMN cell seems to maintain ROS production, and thus, its antimicrobial effect, for longer under normobaric normoxia.

#### 3.2.2. Oxidative Burst after HBO Treatment

FACS analysis after HBO treatment revealed an increased neutrophil oxidative burst for both normobaric and hyperbaric hyperoxia (see Section 2.2.3). Thereby, PMA activation led to a higher respiratory burst than activation with fMLP/TNFα independent of treatment. The observed increased oxidative burst suggests enhanced ROS production at hyperoxia conditions—possibly only by a shift in the chemical reaction equilibrium of ROS production in favor of the product (ROS) because of a higher concentration of the educt (O_2_). Our data agree with observations by Schwartz et al., who found that the addition of HBO during PMN stimulation with PMA results in a significant increase in intracellular ROS production. Moreover, our data agree with Almzaiel et al., who found that a single treatment with hyperbaric oxygen (97.7% O_2_ at 2.35 ATA compared with normoxia, normobaric hyperoxia, and hyperbaric normoxia) postexposure led to an increase in ROS production in neutrophil-like cells [20].

Nonetheless, there are contradictions in the literature as well. Grimberg-Peters et al. reported, that HBO therapy suppresses ROS production in inflammatory human PMNs and thus might impair ROS-dependent pathways, e.g., kinase activation [27].

Interestingly, short-term (5 min) hyperoxia exposure showed increased ROS levels when analyzed immediately afterward in FACS. In live cell imaging, long-term hyperoxia showed no difference in the time of maximum ROS production and even a faster decrease in ROS intensity at the single-cell level. One possible reason for this could be intracellular neutrophil mechanisms or in plasma (used in PMN isolation) that mitigate the effects of hyperoxia during prolonged exposure. ROS act in conjunction with several redox systems involving glutathione, thioredoxin, and pyridine nucleotides and play central roles in coordinating cell signaling as well as protective antioxidant pathways [18]. If cells and plasma are present in the experiments, antioxidant mechanisms seem to work well. In their absence (see the results of the matrix analysis in Section 2.2.4) or if the measurement can take place before the reduction mechanisms are activated, increased ROS levels are found after hyperoxia exposure. In preclinical studies, many of these positive anti-inflammatory effects of both normobaric and hyperbaric oxygen have been repeatedly shown, often as surrogate end-points such as increases in gluthatione levels, reduced lipid peroxidation, and neutrophil activation [18].

In this context, one notable aspect of the cellular response to oxygen fluctuations is the significance of “pulsed” oxygen exposure, as opposed to a singular high or low concentration event. Living subjects typically revert to their baseline oxygen levels post-treatment, a phenomenon often referred to as the “normobaric oxygen paradox”. In this way, the duration of the oxygen treatment in particular plays a crucial role in influencing outcomes at the tissue, cellular, and organismal levels [38].

Nevertheless, our results indicate that HBO increases neutrophil capacity to perform respiratory burst, which points at the importance of HBO on the modulation of the host immune system during infections, as also reported by others [25,39,40]. Moreover, Schwartz et al. reported, that combining HBO with host factors by means of PMNs resulted in substantially decreased bacterial growth, indicating an additive effect of HBO and PMNs [25]. This is accompanied by a direct bacteria-killing effect on anaerobic infections, which has been reported for HBO [41]. Although not revealed thoroughly, the beneficial effect of HBO in the context of severe bacterial infections is suspected to be attributed to direct antimicrobial action, influencing PMNs and enhancing the activity of certain bactericidal antibiotics, probably acting in concert in an in vivo situation [25].

Furthermore, HBO has been demonstrated to improve tissue perfusion, promote angiogenesis, increase the oxygen level in tissues, and inhibit toxin production [41]. As mentioned in the Introduction, many non-healing tissues are hypoxic. Burn therapy, for example, must be directed to minimizing edema, preserving marginally viable tissue, enhancing host defenses, and promoting wound closure. Adjunctive hyperbaric oxygen therapy can attack these problems directly, maintaining microvascular integrity, minimizing edema, and providing the essential substrate necessary to maintain viability [42]. Even though the experimental setups and hyperbaric oxygen treatments in the literature differ from ours, the consistent results further underline an immune-enhancing component of HBO [25].

### 3.3. Neutrophil Surfaces and Oxidative Burst after Pressure Treatment

Since the mid-1980s, it has been known that after a scuba dive, which is considered to be an event of hyperbaric hyperoxia, an elevation of microparticles (in particular, CD66b, CD142, and the von Willebrand factor) in the blood and on cell surfaces occurs in divers. However, this usually returns to normal within 24 h [43]. In our experiments, we also found an influence of pressure and oxygen on the expression of neutrophil surface proteins. PMNs showed an increase in CD11b in normoxia in comparison with the other oxygen conditions. CD62L showed significantly lower levels in normoxia, and CD66b had lower expression in the CAP group than in the HBO and normoxia groups. In 1997, Thom et al. described that HBO exerts an influence on neutrophil adhesion by affecting β2-integrin [44]. Studies to date suggest that HBO exposure increases NO production locally, which may then reduce the adhesive qualities of the PMN via its effects on cGMP synthesis, which presumably affects CD18 function and also reduces the adhesive qualities of endothelial cells via its effects on CAM expression including ICAM-1 [40]. It is even reported that after a stroke, HBO impairs the expression of β2-integrins, and thus, HBO limits cerebral infarct volume and PMN sequestration [40,45]. It is worth thinking that lower expression of the adhesion protein CD11b in HBO conditions, which was observed in our experiments, contributes to this effect. Overall, neutrophil activation by HBO can be detected, and it was considered by Thom et al. as an oxidative stress response to high gas pressure exposures [43].

### 3.4. Determination of H_2_O_2_, Rhodamine 123, NO_2_^−^, and NO_3_^−^ in Different Matrices after Pressure Treatment

Our experiments revealed clearly significant differences in individual reactive metabolites in the tested culture media depending on the pretreatment. H_2_O_2_ was highest in RPMI, followed by DMEM, whereby the difference in the amount of H_2_O_2_ in RPMI compared with the other media was greater than that triggered by pressure and hyperoxia (see Figure 8). ROS (detected by Rhodamin 123 fluorescence) were lowest overall in human plasma. It made no difference whether PMNs were in the plasma or not. These results are consistent with known increases in intracellular and tissue H_2_O_2_ and oxygen radical levels after HBO exposure. Mechanisms within PMNs that protect the cells from increased levels of superoxide, H_2_O_2_, hydroxy radicals, and singlet oxygen could be responsible and include PMNs’ ability to increase the production of superoxide dismutase and glutathione enzymes [26]. In addition, Ferrer et al. reported that after scuba diving, antioxidant enzyme adaptations were upregulated in individuals in order to avoid oxidative damage. They showed that a situation of increased O_2_ consumption led to the activation of antioxidant enzymes such as catalase and glutathione peroxidase, thus supporting a protective role for low dosages of ROS after hyperbaric hyperoxia [46].

In contrast, NO_2_^−^ was highest in human plasma. There was also no difference regardless of whether PMNs were in plasma or not. While there were minor differences in NO^2−^ due to the choice of medium, the differences in NO_3_^−^ between the individual media were considerable. NO_3_^−^ was highest in RPMI, followed by plasma (resp. PMNs in plasma). Our data support reports that scuba diving (which combines hyperoxia and hyperbaric pressure) induces nitrosative damage with increased nitrotyrosine levels and an inflammatory response in PMNs. According to Sureda et al., neutrophil nitrite levels, indicative of inducible nitric oxide synthase (iNOS) activity, progressively increased after diving and recovery. Moreover, NO inhibited neutrophil adherence to the endothelium by inhibiting the expression of adhesion molecules (CD18) following a dive (see Section 2.2.3) [3].

Overall, treatment with CAP significantly increased the amount of RNS and ROS, regardless of the medium used. Treatment with HBO, hyperoxia, or normoxia did not result in significant differences in the respective media in the vast majority of cases. Therefore, when performing hyperbaric experiments, the choice of media in which to perform experiments with hyperbaric and hyperoxic conditions is critical, significant differences in the formation of reactive metabolites are due to the medium only. In particular, the higher amount of H_2_O_2_ in RPMI should be taken into account. In summary, the chosen assay medium can influence the functionality and viability of PMNs and is therefore an important part of an in vitro experiment, which should be carefully selected not only in the context of hyperbaric investigations [25,47].

### 3.5. Features and Limitations of the In Vitro Model

Although the transferability of in vitro results to humans is not always directly possible, such experiments are of great importance in biomedical research. Thereby, research on “normal” and healthy PMN cells is crucial as a foundational step. By studying “normal” cells first, we are able to generate knowledge and establish a baseline for normal cell function for a fundamental understanding of their normal functions and properties. This, in turn, allows us to recognize variants and change patterns in the event of illness. This approach helps to better investigate the causes and mechanisms of diseases and to develop potential treatment approaches [48].

One major challenge for these assays is the isolation of neutrophils from whole blood without activating them or altering their phenotypes. A second critical challenge in studying neutrophil chemotaxis is establishing more complex gradient generation schemes to study the sequential effects of different chemokines or the activity of anti-inflammatory agents. To closely mimic the in vivo scenarios existing around cells, according to Agrawal et al., a simple and efficient in vitro chemotaxis study demands two essential steps as follows: (1) the isolation and immobilization of neutrophils on the device for subsequent microscopy and monitoring cellular responses and (2) the ability to create a dynamic yet controlled and complex environment of multiple chemokines around the cells [49].

#### 3.5.1. Collection and Isolation Methods for Neutrophil Granulocytes

According to Elghetany et al., the choice of anticoagulants in blood collection tubes can have an impact on the individual functions of PMNs [50]. Lithium heparin blood collection tubes were used to collect whole blood in the experiments in this work, as the heparin anticoagulant was found to be the most suitable substance for performing functional PMN assays in a study published by Hodge et al. [51]. Neutrophil granulocytes are sensitive to mechanical stress, which occurs during centrifugation. The high g-forces can impair their structures and functions and lead to activation [24].

The advantages of PMN isolation from whole blood by physical density gradient centrifugation (leuco/lymphospin technique) were primarily the low number of washing steps and the high degree of purity of the isolated cells. Elghetany et al. described the latter as advantageous in terms of avoiding interfering signals when fluorescently labeled antibodies were used in an experimental setup (as performed in our project) [50].

#### 3.5.2. Different Oxygen Partial Pressures that PMNs Encounter

PMNs may encounter a wide range of different O_2_ concentrations [28]. Circulating PMNs come into contact with an oxygen partial pressure of circa 21 kPa (159 mmHg) in the alveoli in purely physiological terms (20.9% oxygen content of the room air at 101 kPa air pressure at sea level), whereas the arterial oxygen partial pressure is usually between 9 and 13 kPa (70 and 100 mmHg) [29]. The sweep gas of an ECMO consists of up to 100% oxygen. Under normobaric conditions, this means that PMNs are exposed to about eight to eleven times the maximum physiological oxygen partial pressures found in the body (in this case, 760 mmHg). Compared with partial pressures in the skeletal muscles used as an example, this is 11 to 63 times higher. This is therefore a remarkably wide range of oxygen concentrations to which granulocytes can potentially be exposed.

#### 3.5.3. Isolated PMNs as the Suitable Analysis Method

In the human body PMNs do not occur isolated and separated; they are always embedded in a network of the extracellular matrix (see Section 3.5.4) and neighbor a large diversity of other (immune) cells. However, it is difficult to distinguish the effects of oxygen on neighboring cells from the confounding effects of neighboring cells on neutrophil function. The sensitivity of the measurement systems used must allow precise conclusions to be drawn about the effects on PMNs, at least because of the wide range of oxygen concentrations, which works best if PMNs can be viewed in isolation. Nevertheless, isolation in itself can also have an impact on PMNs. Centrifugation was used for the isolation of granulocytes and lymphocytes, which, according to Hundhammer et al., could have influenced cell activity [24].

However, the influence of different oxygen concentrations on the function of leukocytes has already been investigated by a few studies. In contrast to the present work, most reports on isolated neutrophils concern bacterial killing, which promoted migration and oxidant production and did not lead to clear results. It has already been established that a single treatment with hyperbaric oxygen (97.7% O_2_ at 2.35 ATA compared with normoxia, normobaric hyperoxia, and hyperbaric normoxia) post-exposure led to an increase in ROS production in neutrophil-like cells [20]. In addition, normobaric hyperoxia increased the phagocytotic activity of the cells. However, Kiers et al. [21] concluded that short-term hyperoxia does not influence phagocytosis or ROS production. Branitzki-Heinemann et al. also stated that hypoxically treated granulocytes produced less ROS after stimulation [22].

#### 3.5.4. Advantages and Limitations of the In Vitro Examination System

The µ-slide chemotaxis chambers^®^ from IBIDI (IBIDI Ltd., Planegg, Germany) used in this work made it possible to embed cells in a matrix so that they were surrounded by the extracellular matrix in a three-dimensional environment, similar to human tissue. Therefore, this model is generally well-suited for the simulation of in vivo conditions of the interstitium [52,53]. Although the µ-slide chambers created stable test conditions over a long period of time, it was basically a static system [54,55]. Interstitial flow, which corresponds to the mechanical changes in human tissue caused by blood flow, was absent. Although the PMNs embedded in a gel matrix were subjected to physical shear forces, these forces differed significantly from the diversity in physiology [56]. However, the biggest limitation of the setup used, with all its advantages, is probably the lack of the following very important step in the PMN migration cascade: the migration of PMNs from the blood vessel through the endothelium to the tissue [56].

Within our model, we attempted to generate stable study conditions in order to obtain reliable measurements by maintaining a constant environment. Care was also taken to keep the oxygen concentration stable for the individual measurements in order to avoid unnecessary bias. However, as explained above, this stable concentration is not physiological. In order to mimic the physiology of changing oxygen concentrations, several series of experiments with different O_2_ concentrations were carried out. A model that combines stable and changing concentrations is difficult to hardly feasible, and it is questionable whether reliable values could be generated at all [49].

The simultaneous LCI test setup made it possible to observe identically prepared samples under identical experimental conditions in two microscopes. Since two climatic chambers were available, control and experimental conditions could be considered simultaneously. Overall, this study, with a partially small sample number, should be seen as a pilot study that can be further supported by larger studies. Nevertheless, the evaluation with simultaneous LCI showed significant differences between the medians of the track lengths in the first 3 h of observation (see Table 1). To ensure accuracy, we decided to analyze this observation period more precisely (in 30 min time slots). During LCI after pressure treatment, TL was very constant over 3 h; significant differences were only seen after 3 h and especially towards the end of the observation period after 6 h (see Figure 6). Therefore, the analysis here was performed in 60 min time slots. This study is an in vitro study, and the in vivo transferability still has to be verified.

#### 3.5.5. Selection of the Analysis Parameters

In previous studies in the literature, leukocytes were examined especially regarding their phagocytosis after incubation with different oxygen conditions. Almzaiel et al.—for example—investigated ROS formation by photometric measurement of the reduction in ferricytochrome c and phagocytosis by incubating neutrophil-like cells with a defined number of *Staphylococcus aureus* bacteria. After incubation, phagocytosis was stopped, and the remaining bacteria were counted by light microscopy [20]. The aim of our study was to investigate the influence of different oxygen concentrations on the function of PMNs. We chose the examination of neutrophil surfaces as indices for neutrophil activation status because the changing of neutrophil surface epitopes is crucial for extravasation from the blood vessels and also so that the subsequent chemotactic migration to the target site can take place [11,33]. Furthermore, we selected ROS production and NETosis as parameters for investigation, as these central neutrophil functions play a key role for PMNs in the fulfillment of their antimicrobial tasks [11].

### 3.6. Comprehensive Evaluation of the Findings

Under normobaric hyper- or hypoxic conditions, PMNs migrate faster and farther, especially at early time slots after the start of observation. It seems that the pressure component would have less effect on the migration behavior of PMNs than the O_2_ concentration in the vicinity of PMNs. Different O_2_ concentrations or pressure conditions to which PMNs are exposed for prolonged periods or treated for short periods do not appear to affect the timing of maximal ROS production or NETosis. However, individual PMN cells produce prolonged ROS at normoxic conditions. PMNs showed a higher expression of CD11b in normoxia, lower expression of CD62L in normoxia, and lower expression of CD66b after HBO and CAP treatment. The choice of nutrient media in which to perform experiments is critical, as there are significant differences in the formation of reactive metabolites solely because of the medium. Treatment with CAP increases the amount of ROS and RNS in commonly used culture media.

Especially the upregulation of CD11b (important for adhesion and transmigration), farther and faster migration (essential for reaching the target of immune action through extracellular tissue), and increased ROS production (necessary for the destruction of pathogens) indicate an increased activation status of PMNs due to higher oxygen levels in the environment of PMNs [57]. The question arises whether PMNs are activated by oxygen itself or by changes triggered by oxygen. A direct influence of oxygen seems likely since O_2_ is directly involved as a reactant in the respiratory burst reaction. In this context, another interesting target of ROS that needs to be discussed is the hypoxia-inducible factor 1α (HIF-1α) [22]. HIF-1α was initially known to act as a transcriptional activator functioning as a master regulator of cellular and systemic oxygen homeostasis. Nowadays, HIF-1α has additionally been shown to play a role in the production of defense factors and to improve the bactericidal activity of myeloid cells. Since HIF-1α is a global regulator of the cellular response preferential to low oxygen and NET formation showed no changes dependent on the oxygen conditions, we suggest that HIF-1α is not solely responsible for the observed reactions. Nevertheless, as already specified by Branitzki et al., our experimental setup with changing oxygen conditions between hypo- and hyperoxia might not be the ideal tool to analyze HIF-1α.

Moreover, recently, a new hypothesis was proposed in the literature that the stability of neutrophil cytosolic oxidation-sensitive proteins may be affected by the transient exposure of PMNs to oxygen (hyperoxia). Two clusters of proteins belonging to cholesterol metabolism and to the complement and coagulation cascade pathways, which are highly susceptible to neutrophil oxygen exposure, were identified. However, the performed pathway enrichment analysis did not reveal any metabolic pathways specifically expressed in neutrophils purified under hyperoxic conditions. In contrast, under anoxic conditions, the lipid and cholesterol metabolism pathways were uniquely enriched. Further investigations are required to address this question, which may lead to an improved understanding of neutrophil physiology under basal conditions and the identification of novel signaling pathways involved in neutrophil activation [58].

Overall, the observed changes in neutrophil function are a normal host defense reaction. However, the mechanisms used by PMNs to kill microorganisms also have the potential to injure healthy tissue. There are several examples (e.g., pneumonia and ARDS) where targeted and supportive PMN activity is useful in disease control but then turns into overwhelming and harmful PMN immune activity [11]. This work does not allow for an assessment of whether the observed changes in function are basically or abnormal. Rather, it depends on the circumstances in vivo when this activation might cause pathology.

The observed effects of activated isolated PMN cells must always be considered in a greater context. As already described, PMNs are always embedded in a network of neighboring cells and the extracellular matrix in vivo. However, in order to understand the neutrophil response, it is essential to look at the behavior of PMNs in isolation. Since there is still no targeted manipulation of PMN and we do not yet understand many neutrophil processes in relation to oxygen, the experimental focus of this study was on PMN responses in relation to oxygen. From our point of view, we cannot determine that one part outweighs the other. Both parts must be considered independently of each other first and finally assessed together again under consideration of the context.

In summary, our data show an increased neutrophil activation shortly after the application of hyperoxia expressed by increased neutrophil ROS production, upregulated CD11b, and downregulates CD66b in the FACS analysis and increased migration in the chemotaxis experiment. These mechanisms and, especially, the cumulative release of oxidative stress (ROS), together with delayed or absent activation of antioxidant mechanisms (which we found in different matrices), could explain why, in the absence of hypoxemia, liberal use of oxygen therapy in critically ill patients showed no benefit but rather appeared to be harmful. This increased neutrophil activation may be why hyperoxia can have serious negative effects on pre-existing inflammation and oxidative stress in patients with myocardial infarction, stroke, traumatic brain injury, cardiac arrest, and sepsis [12].

## 4. Materials and Methods

The workflow of the experiments is shown in Figure 1. We performed live cell imaging (LCI) chemotactic experiments and determined neutrophil migration, ROS production, NETosis, and changes in surface epitopes as a measure of neutrophil activation. Thereby, two different experimental setups were performed as follows: simultaneous LCI (Figure 1a) and LCI after pressure treatment (Figure 1b). In addition, the influence of different O_2_ conditions on nutrient media (RPMI, MEM, H_2_O, blood plasma) was investigated by ROS and RNS measurements (Figure 1c).

### 4.1. Approval of the Ethics Committee and Selection of Study Participants

This study was conducted in accordance with the Declaration of Helsinki and approved by the Ethics Committee of the University Hospital Regensburg (file number: 16-101-0322). Healthy nonpregnant human volunteers, who gave informed consent, were included in this study. Each experiment was performed with different individual test persons so that the number of experiments (*n*) could be equated with the number of test persons. Males and females were represented similarly in the test population, whose age range was from 20 to 55 (with an average of 28.3 years). At the time of blood sampling, participants neither showed symptoms of an infection nor had an acute or chronic PMN-affecting disease.

### 4.2. Sample Collection and Preparation

Granulocytes were isolated from lithium heparin-anticoagulated whole blood of healthy donors by means of density gradient centrifugation. At ambient temperature, centrifugation was performed for 20 min at 756× *g* through PBMC (Lympho) Spin Medium on top of Leuko Spin Medium (both pluriSelect Life Science, Leipzig, Germany). After centrifugation, various macroscopically well-defined layers were obtained, each containing different cell types. The layers above the granulocytes were aspirated using a water jet pump. The granulocyte ring, which was now directly accessible, was separated and washed once.

The further use of the isolated cells was then dependent on the respective tests, as shown in Figure 1.

For simultaneous live cell imaging (see Section 2.1), they were resuspended in Roswell Park Memorial Institute (RPMI)-1640 culture medium (Pan Biotech Ltd., Aidenbach, Germany) and 10% fetal calf serum (FCS, Sigma-Aldrich, Taufkirchen, Germany) at a concentration of 18 × 10^6^ cells/mL, as reported in detail before [33,35], and further processed as described in Section 4.5 and Section 4.6. For pressure treatment (see Section 4.3), the cells isolated by density gradient centrifugation (see above) were first resuspended in autologous plasma after centrifugation. To obtain autologous plasma, the above-mentioned lithium heparin-anticoagulated blood was centrifuged for 10 min at 1.181× *g* (21 °C) (Heraeus Megafuge type 1.0 R, Thermo Fisher Scientific, Waltham, MA, USA). Subsequently, the supernatant plasma was decanted and further used [24]. The PMN cells resuspended in autologous plasma were processed further as describe in Section 4.4. For live cell imaging after pressure treatment, PMNs isolated by density gradient centrifugation (see above) were resuspended again to a concentration of 18 × 10^6^ cells/mL with RPMI (+10% FCS) analogously to the procedure above (see Section 4.2) and processed further as described in Section 4.5 and Section 4.6.

### 4.3. Provision of the Individual Matrices for Pressure Treatment

In order to prepare the selected matrices for the pressure treatment, 2.5 mL of the respective matrices (see Table 5) were filled into 35 mm Petri dishes (Corning, Merck, Darmstadt, Germany). Each of the 4 conditions (see Table 5) was linked with each of the 5 matrices. Every combination (see Table 5) was duplicated, resulting in (2 × 5 × 4 = 40) Petri dishes filled for pressure treatment and subsequent analysis.

### 4.4. Pressure Treatment

A total of *n* = 3 subjects performed the pressure treatment. The provided dishes (see Section 4.3) were treated in the pressure chamber HAUX Testcom 200/2 (Haux-Life-support GmbH, Karlsbad-Ittersbach, Germany, see Figure 9) for 90 min each at 35–37 °C according to the conditions listed in Table 6.

The application of cold atmospheric plasma (CAP) to normoxic and normobaric culture media served as positive control. For treatment with (CAP), a prototype of the plasma care^®^ device (Terraplasma GmbH, Garching, Germany), a surface micro-discharge (SMD) plasma source, was used. The unit consists of a high-voltage electrode and dielectric and grounded structured electrodes, which subsequently produce plasma components alterable by frequency and voltage. The usage of a high voltage of 3.5 kV provokes millimeter-sized micro-discharges into a plasma source unit. By using a frequency of 4 kHz (“oxygen mode”), the plasma care^®^ device produced Ozone (O_3_) in parts per million (ppm). For condition d) (see Table 6), 2.5 mL of the corresponding matrix (see Table 5) was treated for 5 min with a frequency of 4 kHz and a duration of 5 min, as described before by Kupke et al. [59].

Subsequently, the samples were submitted to the respective analytical methods (see Figure 1 and also Section 4.5, Section 4.6 and Section 4.8).

### 4.5. Migratory Chemotaxis Model

First, 24 h prior to each observation, the µ-Slide chemotaxis (IBIDI-Slide, IBIDI Ltd., Planegg, Germany) was stored in an incubator at 37 °C and 5% CO_2_. Each IBIDI-Slide consists of three chambers, with each chamber consisting of a central channel and a reservoir on each side. The cell medium was prepared in advance using 60 µL Minimum Essential Media (MEM, Pan Biotech), 60 µL H_2_O, 30 µL sodium hydrogen carbonate (1 M, NaHCO_3_, VWR International Ltd., Radnor, PA, USA), and 150 µL RPMI-1640 and kept in the incubator next to the IBIDI-Slide. The PureCol Type I Bovine Collagen Solution (collagen, 3 mg/mL, Advanced BioMatrix Carlsbad, CA, USA) was at room temperature when used. The IBIDI-Slide was then incubated for 30 min at 37 °C and 5% CO_2_.

Afterwards, the right reservoir of each chamber was filled with 65 µL of RPMI 1640 and 10% FCS, whereas the left reservoir contained 65 µL of fMLP (10 nM in RPMI/10% FCS) [33,34,35]. The final prepared samples were then exposed to the experimental gas in the microscope´s climate chamber, and the observation procedure described below was started.

### 4.6. Microscopy and Live Cell Imaging

Migration and fluorescence were measured according to Doblinger et al. using a DMi8 inversion microscope (Leica Microsystems, Wetzlar, Germany) equipped with an IBIDI stage top incubator (37 °C, 5% CO_2_), and photographic images were obtained using a DFC9000 GT SCMOS black and white camera (Leica Microsystems), a CoolLEDpE4000 light source (CoolLED Ltd., Andover, Great Britain, UK), and Leica Application Suite X (10) Software (LAS X (10) 30,416,529, Leica Mikroskopie & Systeme Ltd., Wetzlar, Germany) [35]. The control of the microscope and the observation process were computer-aided using the Application Suite X (10) software platform (Leica Microsystems) [33,35]. Photos were taken at 30 s time intervals for a total duration of 330 min. Migration was tracked in phase contrast. ROS production was assessed by means of 1 µM dihydrorhodamine 123 (DHR; Thermo Fisher Scientific). NETosis was visualized using 5 µM 4,6-diamidino-2-phenylindole (DAPI; Sigma-Aldrich) and MPO release using 0.5 µg/mL anti-MPO-APC antibodies (Miltenyi Biotech, Bergisch Gladbach, Germany). The detailed settings of the fluorescence filters are listed Table A1 in Appendix B [33].

The image series generated by the microscope were analyzed using Imaris^®^ 9.0.2 computer software (Bitplane AG, Zurich, Switzerland, see Figure 10 and Appendix A). To quantify migration, axes in the x and y directions were defined for each channel, as in a Cartesian coordinate system, over the entire length of the considered image area so that defined migration quantities could be determined (see Table A2 in Appendix C) [33]. For statistical analysis, the filters “TrackDuration above 900 s” and “TrackLength > 25 µm” were set to exclude artifacts. The results of the migration analysis were subdivided into 30 min (simultaneous LCI) and 60 min (LCI after pressure treatment) observation intervals (time slots). The determined migration time slots were set from the start of the microscopic observation.

ROS production was assessed by determining the time of maximum ROS production (TmaxROS) using Excel 2021 (Microsoft Corporation, Redmond, Washington, DC, USA) as described previously [35].

To establish the average level of ROS production of single neutrophil cells over 6 h, Rhodamine 123 fluorescence intensities were quantified by detecting the median fluorescence intensity of each cell in every microscope image (Imaris parameter “IntensityMedianROS”). By calculating the median of the “IntensityMedianROS” of all detected cells, the Median intensity of the Rhodamin 123 fluorescence response was determined per image and then called the “Relative Fluorescence Index” (RFI). The values of RFI were plotted against the time points divided and sorted into ten-minute intervals. Statistical analysis (see Section 4.10) was then performed by comparing the values at time points t = 10 min (10 min after the start of the experiment), t = 30 min, and t = 60 min. NETosis was evaluated by determining the time of the half-maximal effect size (ET_50_NETosis) using Phoenix^®^ 8.0.0 (Certara L.P., Radnor, Pennsylvania) as described before [33,35].

### 4.7. Distinctive Feature of Simultaneous Live Cell Imaging

For the distinctive feature of simultaneous live cell imaging, a simultaneous experimental procedure with identically prepared samples was viewed simultaneously under identical experimental conditions (except O_2_ concentration; see below) in two nearly identical microscopes. For simultaneous live cell imaging, isolated PMNs were not pre-treated in the pressure chamber. The variation in the O_2_ concentration was performed during the microscope observation (see below). Microscopic observation and evaluation were performed as described in Section 4.6. The simultaneous live cell imaging procedure is shown in Figure 11.

A total of *n* = 10 experiments were performed with PMNs from three different subjects each in normobaric conditions with 1 ATA pressure. The O_2_ concentration of 4% with an oxygen partial pressure of 30.4 mmHg was classified as “normobaric hypoxia”. The O_2_ concentrations of 50% (oxygen partial pressure: 380 mmHg) and 83% (partial pressure: 631 mmHg) were declared as “normobaric hyperoxia”. As a control, all experimental series were performed in pairs with simultaneous comparison to the 20% oxygen concentration (control with oxygen partial pressure of 152 mmHg). For this control, 95% compressed air as well as 5% CO_2_ flowed through the climatic chamber at 50% relative humidity.

For changes in oxygen concentrations, the anesthesia machine Trajan 808 (Dräger, Lübeck, Germany) in the experimental setup shown in Figure 12 was used. By mixing 5% CO_2_ with 95% test gas, which was in turn mixed in an IBIDI Gas Mixer (IBIDI Ltd., Planegg, Germany), O_2_ concentrations between 4% and 83% could be created. A schematic illustration of the gas flow and the connection of the anesthetic machine is shown in Figure 12.

### 4.8. Measurement of Cell Surface Antigens and Respiratory Burst by Flow Cytometry after Pressure and Hyperoxia Treatment

Flow cytometry (FACSCalibur, BD Biosciences, San Jose, CA, USA) was applied along with CellQuest Pro software (version 5.2, BD Biosciences, Franklin Lakes, NJ, USA) to assess cell surface antigen expression and respiratory burst. Cell surface antigen expression was detected by means of the fluorochrome-conjugated antibodies CD11b (ICRF44, PE-conjugated, BioLegend, San Diego, CA, USA), CD62L (DREG-56, FITC conjugated, BioLegend), and CD66b (G10F5, APC conjugated, BioLegend) as previously described [59]. Oxidative burst was investigated by flow cytometry directly after the neutrophil isolation process. Samples were withdrawn from the leukocyte-rich supernatants gained in the isolation process that were not further treated. Burst was stimulated by Formyl–Methionin–Leucin–Phenylalanin (fMLP, 10 µM, Sigma-Aldrich) in combination with Tumor necrosis factor α (TNFα, 1 µg/mL, Thermo Fisher), or otherwise with Phorbol-12–myristat-13–acetat (PMA, 10 µM, Sigma-Aldrich), which was set as the positive control [24]. Neutrophil respiratory burst was measured in 20 µL of pretreated (see Table 5) cells suspended in 1 mL of Dulbecco’s phosphate-buffered saline (DPBS, Sigma-Aldrich), 10 µL DHR 123 (10 µM), and 10 µL seminaphtharhodafluor (SNARF) fluorescent dye (10 µM, Invitrogen (Thermo Fisher), Waltham, MA, USA). This process has already been described elsewhere in more detail [24,35]. The data analysis was implemented using FlowJo software (version 10.7.1, FlowJo LLC, Ashland, OR, USA) [59]. Thereby, cell counts of 5.000 cells were investigated per sample. The expression of the different surface epitopes by the median fluorescence intensities (MFIs) was compared [35].

### 4.9. Determination of H_2_O_2_, Rhodamine 123, NO_2_^−^, and NO_3_^−^

For ROS detection, 10 µM dihydrorhodamine 123 (DHR123; Sigma-Aldrich) was solubilized in 100 µL of matrices immediately after treatment (see Table 5) and transferred into a black 96-well plate (Greiner-Bio-One GmbH, Frickenhausen, Germany). Subsequently, Rhodamin 123 fluorescence was measured at Ex = 505 nm and Em = 534 nm according to Kupke et al. [59]. H_2_O_2_ was quantified with a Fluorimetric Hydrogen Peroxide Assay Kit (Sigma-Aldrich), and fluorescence was measured at Ex = 540 nm and Em = 590 nm. NO_2_^−^ and NO_3_^−^ concentrations were determined by using the colorimetric Nitrite/Nitrate Assay Kit (Sigma-Aldrich) to detect nitric oxide metabolites at 540 nm absorbance. Light absorption and fluorescence were measured with a Varioscan Flash (Thermo Fisher) [59].

### 4.10. Statistical Analysis

Statistical analyses not conducted in Microsoft Excel 2016 were carried out using SPSS Statistics Version 29 (IBM) [35]. Normal distributions were verified by the Kolmogorov–Smirnov test. For a normal distribution and in the case of multiple comparisons and existing homogeneity of variance, the mean values (MVs) were checked for significant differences by a single-factor analysis of variance (ANOVA) with an indication of the standard deviation ± SD. If there was no homogeneity of variance, Welch’s ANOVA test was applied. In the case of homogeneity of variance, a subsequent post hoc analysis was performed according to Bonferroni. In case of inhomogeneity of variance, Dunnet′s T3 test was used. The test for homogeneity of variance was conducted using Levene’s test. The distribution of results was visualized either with an ATA chart diagram displaying mean values with confidence interval or with simple or grouped boxplots displaying the median, 25% and 75% quartiles, and the calculated minima and maxima. Statistical outliers were represented as circles, and extreme values were depicted as asterisks. If there was no normal distribution in the data to be compared, the central tendencies of the individual groups were compared with Kruskal–Wallis testing, and the median and the interquartile range were reported. The post hoc analysis after this test was conducted by Dunn–Bonferroni testing. An error probability of *p* < 0.05 was considered statistically significant [33].

## 5. Conclusions

Oxygen was and still is often used relatively uncritically in everyday clinical practice. But oxygen, like other drugs, is a drug with useful effects and side effects that must be considered. In the present study, relevant effects at the cellular level that might cause clinical side effects were demonstrated. Therefore, our findings suggest that we have to be aware of adverse oxygen effects not only because hyperoxia can increase neutrophil activation and thus inflammation. Nevertheless, the administration of oxygen therapy remains controversial [27]. Therefore, any form of oxygen application should be subject to critical indication and evaluation.

## Figures and Tables

**Figure 1 biomolecules-14-01242-f001:**
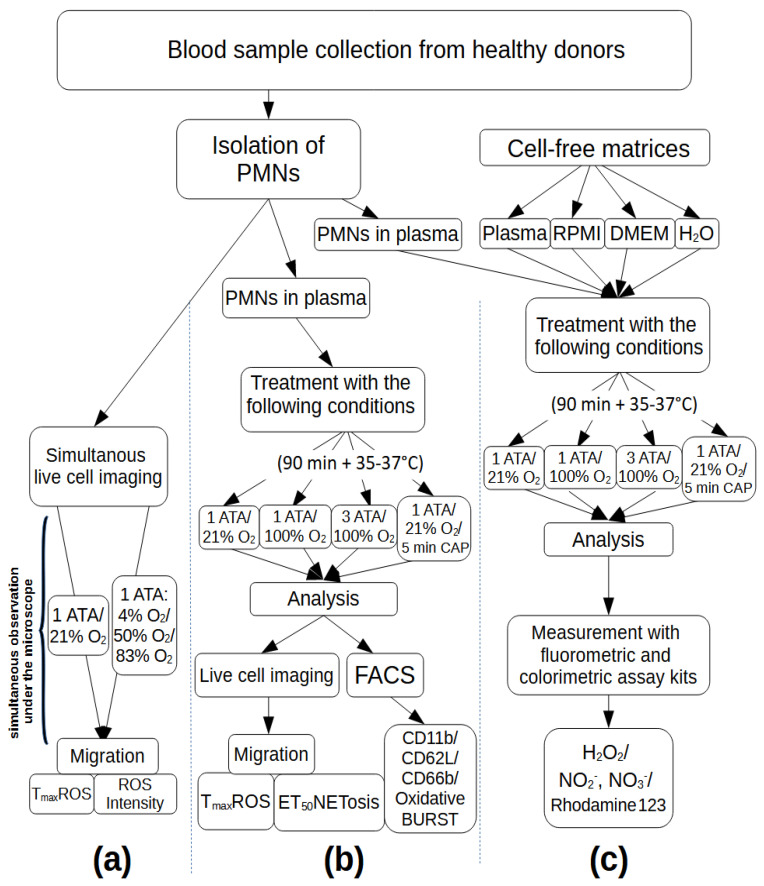
Schematic presentation of the experimental workflow. (**a**) Isolated PMNs were observed for a long time period with distinct oxygen fractures at normobaric pressure levels. (**b**) Isolated PMNs were treated with normobaric and hyperbaric hyperoxia and cold atmospheric plasma (CAP) as a control, whereby neutrophil function and surface epitopes were analyzed by live cell imaging and FACS analysis. (**c**) Cell-free matrices and isolated PMNs in plasma were treated with normobaric and hyperbaric hyperoxia and (CAP) as control. The properties of the media were tested for reactive oxygen and nitrogen species with fluorescence assay kits.

**Figure 2 biomolecules-14-01242-f002:**
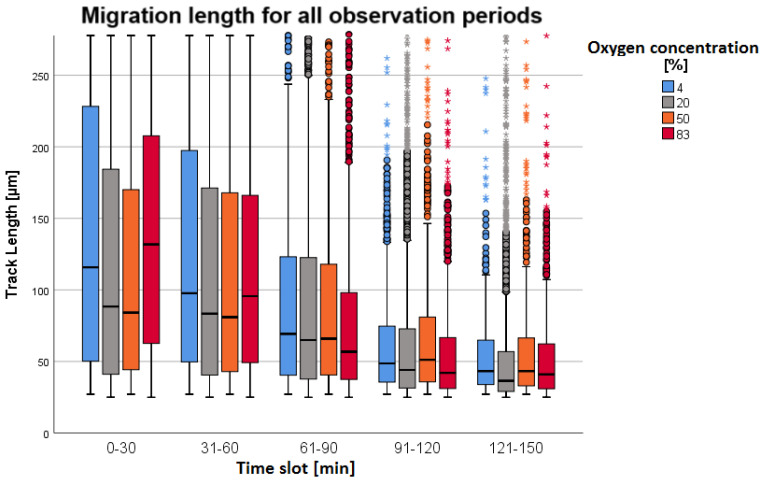
Results of the chemotactic migration analysis of simultaneous live cell imaging. Overview of migration distances of PMNs (TrackLength [µm]), split into observation time slots of 30 min, with data grouped by oxygen concentration. Each boxplot contains the migration routes (tracks) of individual cells. Statistical outliers were represented as circles, and extreme values were depicted as asterisks. The number of cells varied between *n* = 8038 tracks in the first section and *n* = 761 in the last section. Thereby, each test person contributed an average of 500 cells per measurement. In all groups, TrackLength was initially largest and decreased in later observation periods (time slots). PMNs migrated the farthest and fastest in hypoxia (4% O_2_).

**Figure 3 biomolecules-14-01242-f003:**
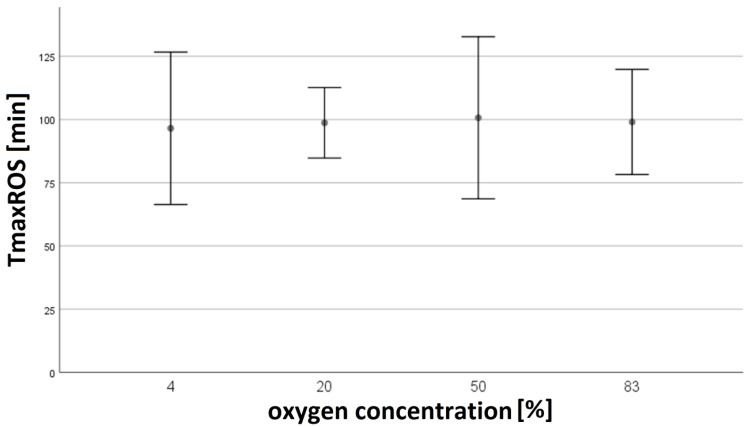
The time of maximum ROS production (T_max_ROS) was determined by live cell imaging. Data is shown as means with standard deviations of T_max_ROS and distinguished by oxygen concentration. The number of experiments varied between *n* = 8 and *n* = 24. No significant differences in T_max_ROS could be observed.

**Figure 4 biomolecules-14-01242-f004:**
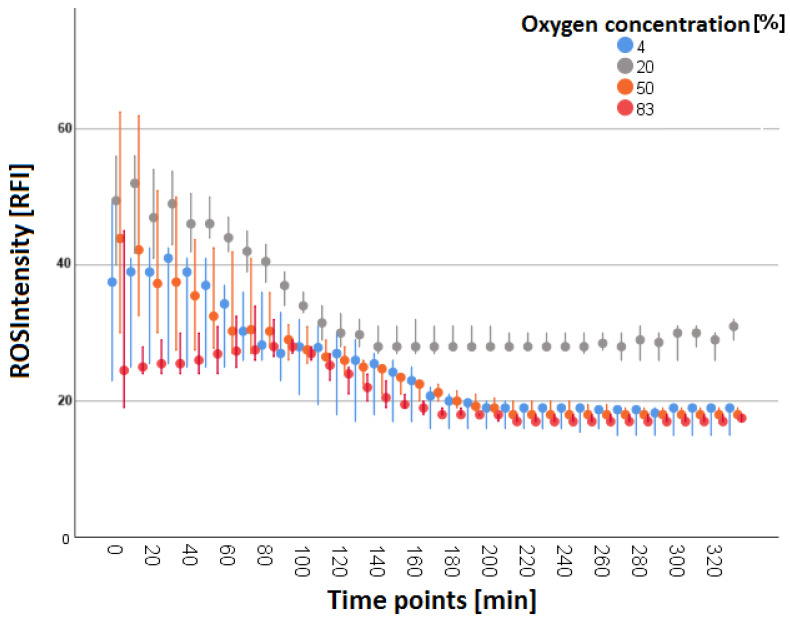
Median RFIs of ROS intensity shown as a dot plot diagram. The *Y*-axis shows RFI, whereas the *X*-axis shows the timeline of the experiment distinguished in 10 min observation time points. The time points are plotted from the start of the microscopy procedure. Data are grouped by oxygen concentration. In all groups, initially, higher intensities were measured, which decreased with time. In normobaric normoxia (20% O_2_), this decrease reached a higher plateau after about 140 min, while in the other groups, it kept falling from the beginning to about 200 min.

**Figure 5 biomolecules-14-01242-f005:**
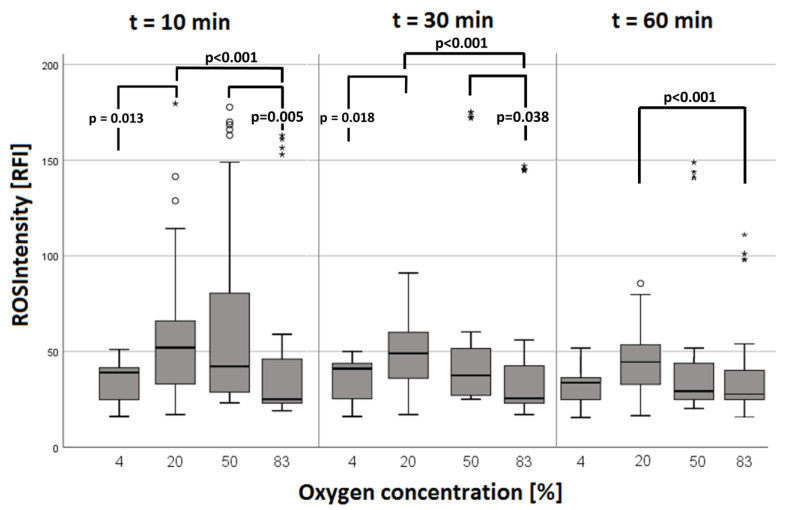
Results of the “ROS Intensity” parameter depending on oxygen concentrations at the t = 10 min, t = 30 min, and t = 60 min time points. A decrease in ROS intensity at a high oxygen concentration of 83% is recognizable. Data are shown as medians with boxes (interquartile ranges) and whiskers (minimum and maximum values); statistical outliers are represented as circles, and extreme values are depicted as asterisks.

**Figure 6 biomolecules-14-01242-f006:**
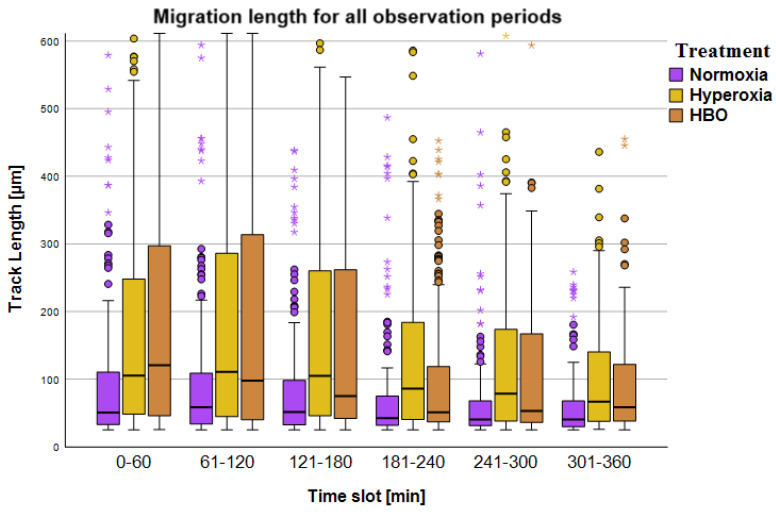
Results of the migration analysis of live cell imaging after pressure treatment. Overview of migration distances of PMNs (TrackLength [µm]), split into observation time slots of 60 min, with data grouped by treatment conditions. Each boxplot contains the migration routes (tracks) of individual cells. The number of cells varied between *n* = 248 tracks and *n* = 144. Thereby, each test person contributed an average of 300 cells per measurement. Data is shown as medians with boxes (interquartile ranges) and whiskers (minimum and maximum values); statistical outliers are represented as circles, and extreme values are depicted as asterisks. TrackLength was significantly lowest in the normoxia group in all time slots. There were little differences between normobaric hyperoxia and HBO.

**Figure 7 biomolecules-14-01242-f007:**
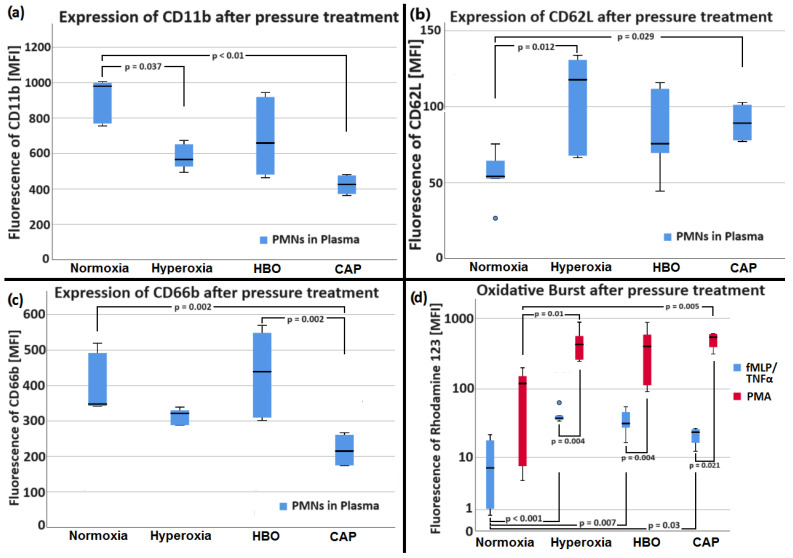
Results of the FACS analysis of the neutrophil surface epitopes CD11b (**a**), CD62L (**b**), and CD66b (**c**) and of the oxidative burst (**d**) after pressure treatment. Data is shown as medians with boxes (interquartile ranges) and whiskers (minimum and maximum values); statistical outliers are represented as circles. PMNs showed an increase in CD11b in normoxia in comparison with the other oxygen conditions. CD62L showed significantly lower levels in normoxia, and CD66b had lower expression in the CAP group than in the HBO group and in the normoxia group. Oxidative burst was higher in the CAP group than in the hyperoxia and normoxia groups.

**Figure 8 biomolecules-14-01242-f008:**
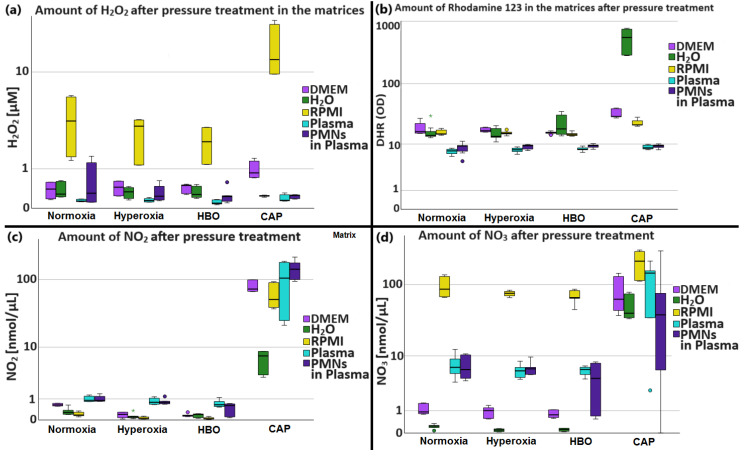
Results of the H_2_O_2_, Rhodamine 123, NO^2−^, and NO_3_^−^ measurements for the different matrices after pressure treatment (*x*-axis). The sections show the different parameters including (**a**) H_2_O_2_, (**b**) Rhodamine 123, (**c**) NO_2_, and (**d**) NO_3_. The cluster variable distinguishes the different matrices. Data is shown as medians with boxes (interquartile ranges) and whiskers (minimum and maximum values); Statistical outliers are represented as circles, and extreme values were depicted as asterisks. H_2_O_2_ was highest in RPMI, followed by DMEM, whereby the difference in the amount of H_2_O_2_ in RPMI compared with the other media was greater than that triggered by pressure and hyperoxia. ROS were lowest overall in human plasma. It made no difference whether PMNs were in the plasma or not.

**Figure 9 biomolecules-14-01242-f009:**
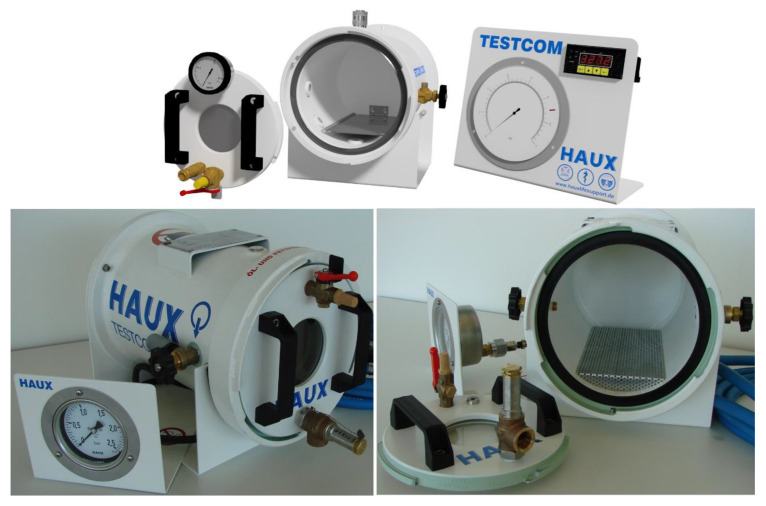
The HAUX Testcom 200/2 pressure chamber used to treat the samples (graphic provided by the Haux-Life-support GmbH).

**Figure 10 biomolecules-14-01242-f010:**
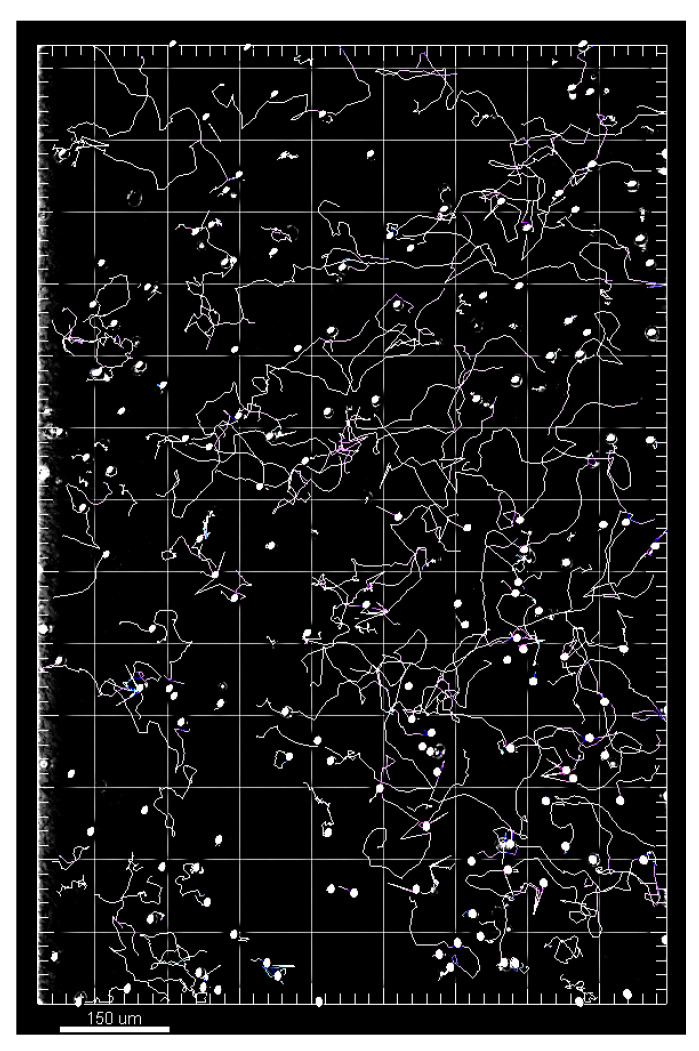
Representation of the migration evaluation by spots (white spheres are detected single PMN cells) and tracks (pink and white lines are tracked and recorded pathways of individual migrating PMN cells) in Imaris.

**Figure 11 biomolecules-14-01242-f011:**
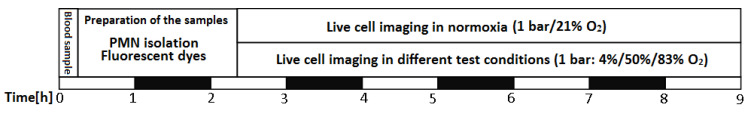
Flowchart of the workflow of the simultaneous live cell imaging with two microscopes. The time axis is plotted to the right, and the steps of the experiment are shown above it. Simultaneously performed steps are shown as vertically layered. Black and white stripes symbolize 1 h observation time each.

**Figure 12 biomolecules-14-01242-f012:**
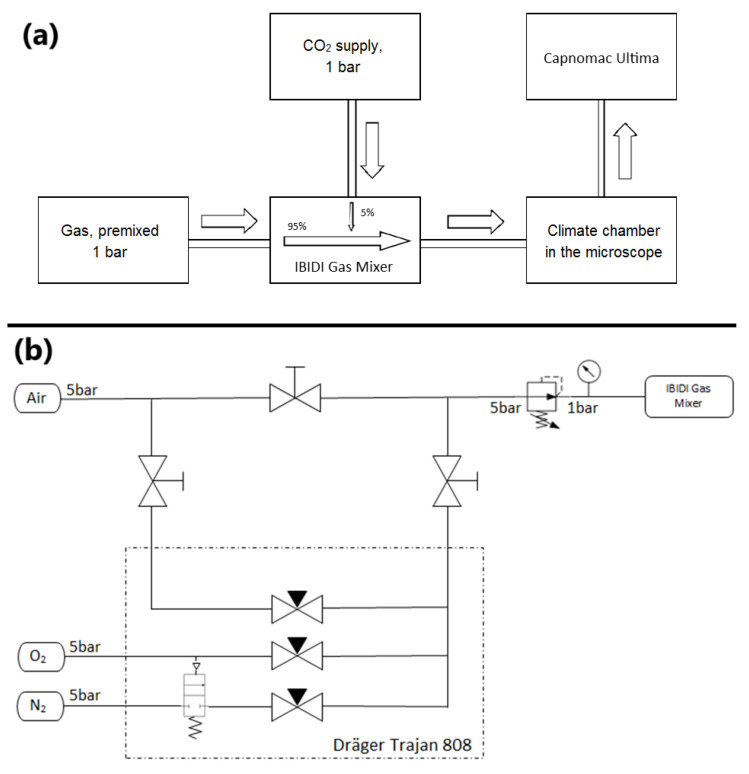
(**a**) Gas flow diagram of the premixed test gas in the climate chamber. The premixed gas was mixed from nitrogen and oxygen depending on the required oxygen concentration. (**b**) Connection sketch of the anesthesia device. On the left side, gas supplies are shown; in the upper part of the picture, the stopcocks for setting the flow path are shown; in the lower part of the picture, the anesthesia machine (for gas mixing) is shown framed; and on the right side, the pressure regulator and the flow path to the downstream IBIDI gas mixer are shown.

**Table 1 biomolecules-14-01242-t001:** Overview of medians and interquartile ranges of the migration lengths (TrackLength) of the different oxygen concentrations in 30 min time slots, which could be observed by means of live cell imaging. (Data is shown as medians with interquartile ranges. In each cell, the uppermost value without brackets indicates the median, the value in round brackets indicates the interquartile range, and the value in square brackets indicates the number of recorded tracks.)

O_2_ Concentration	0–30	31–60	61–90	91–120	121–150	Time Slot [min]
4%	116	97.7	69.3	48.6	43.3	TrackLength/(IQR) [µm] TrackNumber *n*
	(178)	(148)	(82.8)	(39.1)	(31.0)	
	[*n* = 2145]	[*n* = 1713]	[*n* = 1214]	[*n* = 770]	[*n* = 482]	
20%	88.4	83.4	65.0	44.0	36.5	
	(143)	(131)	(84.9)	(41.3)	(27.8)	
	[*n* = 8038]	[*n* = 6698]	[*n* = 5740]	[*n* = 4639]	[*n* = 3984]	
50%	84.1	80.9	66.0	51.2	43.2	
	(126.0)	(125.0)	(77.5)	(45.7)	(33.5)	
	[*n* = 2189]	[*n* = 1715]	[*n* = 1242]	[*n* = 841]	[*n* = 579]	
83%	131.8	95.6	56.7	42.0	41.0	
	(145,3)	(117.0)	(60.8)	(35.7)	(31.4)	
	[*n* = 2541]	[*n* = 2091]	[*n* = 1613]	[*n* = 1073]	[*n* = 761]	

**Table 2 biomolecules-14-01242-t002:** Overview of the medians and interquartile ranges of migration lengths (TrackLength) after pressure treatment in 60 min time slots (first line), which could be observed by means of live cell imaging. (Data are shown as medians with interquartile ranges. In each cell, the uppermost value without brackets indicates the median, the value in round brackets indicates the interquartile range, and the value in square brackets indicates the number of recorded tracks.)

Treatment	0–60	61–120	121–180	181–240	241–300	301–360
Normoxia	50.6 µm	58.2 µm	51.2 µm	42.3 µm)	40.3 µm	40.3 µm
	(77.8 µm)	(75.6 µm)	(66.7 µm)	(44.0 µm)	(37.1 µm)	(38.6 µm)
	[*n* = 161]	[*n* = 176]	[*n* = 150]	[*n* = 180]	[*n* = 204]	[*n* = 180]
Hyperoxia	105.2 µm	110.9 µm	104.7 µm	86.0 µm	78.4 µm	66.7 µm
	(202.3 µm)	(242.9 µm)	(214.5 µm)	(144.4 µm)	(135.6 µm)	(103.7 µm)
	[*n* = 201]	[*n* = 177]	[*n* = 178]	[*n* = 240]	[*n* = 248]	[*n* = 210]
HBO	120.4 µm	97.7 µm	74.9 µm	50.9 µm	53.1 µm	58.3 µm
	(253.4 µm)	(274.2 µm)	(223.0 µm)	(89.5 µm)	(131.4 µm)	(84.8 µm)
	[*n* = 152]	[*n* = 165]	[*n* = 144]	[*n* = 177]	[*n* = 182]	[*n* = 188]

**Table 3 biomolecules-14-01242-t003:** Overview of medians and interquartile ranges of H_2_O_2_, Rhodamine 123, NO_2_^−^, and NO_3_^−^ measurement determinations after pressure treatment. The sections show (1) H_2_O_2_, (2) ROS (Rhodamine 123), (3) NO_2_^−^, and (4) NO_3_^−^.

**(1) H_2_O_2_ [µM] (*n* = 6)**				
Condition	Normoxia	Hyperoxia	HBO	CAP
DMEM	0.403 (0.406)	0.449 (0.370)	0.495 (0.223)	0.863 (0.611)
H_2_O	0.283 (0.387)	0.334 (0.267)	0.275 (0.278)	0.244 (0.251)
RPMI	3.650 (4.699)	3.240 (2.596)	2.225 (1.978)	12.690 (15.297)
Plasma	0.133 (0.049)	0.143 (0.075)	0.093 (0.087)	0.149 (0.140)
PMNs in Plasma	0.313 (1,18)	0.236 (0.358)	0.228 (0.207)	0.252 (0.084)
**(2) Rhodamine 123 [OD] (*n* = 9)**				
Condition	Normoxia	Hyperoxia	HBO	CAP
DMEM	16.63 (8.58)	16.94 (2.96)	15.49 (1.23)	29.23 (10.66)
H_2_O	13.97 (4.16)	13.61 (5.72)	18.16 (18.03)	548.74 (461.54)
RPMI	14.94 (2.99)	15.26 (1.35)	14.57 (1.34)	21.21 (5.66)
Plasma	7.54 (1.70)	7.88 (1,49)	8.01 (1.06)	8.55 (1.54)
PMNs in Plasma	9.21 (2.27)	9.32 (1.56)	9.35 (1.21)	9.26 (1.04)
**(3) NO_2_^−^ [nmol/µL] (*n* = 6)**				
Condition	Normoxia	Hyperoxia	HBO	CAP
DMEM	0.64 (0.14)	0.20 (0.23)	0.16 (0.07)	73.03 (31.70)
H_2_O	0.28 (0.24)	0.10 (0.12)	0.20 (0.15)	7.30 (5.22)
RPMI	0.17 (0.16)	0.05 (0.08)	0.04 (0.06)	51.41 (51.91)
Plasma	0.90 (0.37)	0.79 (0.40)	0.64 (0.35)	104.71 (153.99)
PMNs in Plasma	0.90 (0.37)	0.78 (0.23)	0.60 (0.60)	142.79 (84.45)
**(4) NO_3_^−^ [nmol/µL] (*n* = 6)**				
Condition	Normoxia	Hyperoxia	HBO	CAP
DMEM	0.93 (0.71)	1.01 (0.66)	0.76 (0.45)	64.93 (89.89)
H_2_O	0.20 (0.17)	0.10 (0.10)	0.11 (0.11)	40.65 (20.30)
RPMI	86.84 (62.61)	75.90 (13.87)	66.28 (23.84)	211.50 (171.23)
Plasma	6.81 (4.87)	5.92 (2.53)	6.21 (1.85)	143.14 (140.32)
PMNs in Plasma	6.21 (6.05)	6.35 (2.24)	4.54 (7.19)	46.18 (124.13)

**Table 4 biomolecules-14-01242-t004:** Overview of oxygen effects on PMNs divided into normobaric and hyperbaric treatment modes and hypoxic, normoxic, and hyperoxic conditions. Abbreviations: NB = normobaric; HB = hyperbaric; O_2_ ↓ = hypoxia; O_2_ ↔ = normoxia; O_2_ ↑ = hyperoxia.

Analytical Method	Simultaneous Live Cell Imaging			Live Cell Imaging after Pressure Treatment			FACS				Fluorimetric and Calorimetric Assay Kits			
Pressure	NB			NB		HB	NB		HB		NB		HB	
Oxygen condition	O_2_	O_2_	O_2_	O_2_	O_2_	O_2_	O_2_	O_2_	O_2_	CAP	O_2_	O_2_	O_2_	CAP
	↓	↔	↑	↔	↑	↑	↔	↑	↑		↔	↑	↑	
Migration	↑	↔	↑	↔	↑	↑	-	-	-	-	-	-	-	-
T_max_ROS	↔	↔	↔	↔	↔	↔	-	-	-	-	-	-	-	-
ROS Intensity	↓	↑	↓	-	-	-	-	-	-	-	-	-	-	-
NETosis	-	-	-	↔	↔	↔	-	-	-	-	-	-	-	-
CD11b	-	-	-	-	-	-	↔	↓	↔	↓	-	-	-	-
CD62L	-	-	-	-	-	-	↔	↑	↔	↑	-	-	-	-
CD66b	-	-	-	-	-	-	↔	↔	(↑)	↓	-	-	-	-
* Rhodamine	-	-	-	-	-	-	↔	↑	↑	↑	↔	↔	↔	(↑)
* H_2_O_2_	-	-	-	-	-	-	-	-	-	-	↔	↔	↔	(↑)
* NO_2_^−^	-	-	-	-	-	-	-	-	-	-	↔	↔	↔	↑
* NO_3_^−^	-	-	-	-	-	-	-	-	-	-	↔	↔	↔	(↑)

Note: ↑/↓ indicates significant change, ↔ indicates no change (significance level, *p* < 0.05). () = significance was not achieved for all groups. * A basic statement was made for the trend in all matrices.

**Table 5 biomolecules-14-01242-t005:** Matrices used for pressure treatment. Abbreviations: DMEM, Dulbecco’s Modified Eagle Medium (DMEM, Sigma-Aldrich), RPMI-1640, Roswell Park Memorial Institute 1640 medium. (RPMI-1640, Pan Biotech), Human plasma and PMNs in human plasma (see Section 4.2). Treatment conditions: (a) = normoxia, (b) = hyperoxia; (c) = hyberbaric oxygen (HBO); (d) = cold atmospheric plasma (CAP). For more details of the treatment conditions, see Table 6.

Number	Treatment	Volume in 35 mm Petri Dish [mL]	Matrix
1	(a)	2.5	Distilled H_2_O
2	(b)	2.5	Distilled H_2_O
3	(c)	2.5	Distilled H_2_O
4	(d)	2.5	Distilled H_2_O
5	(a)	2.5	DMEM
6	(b)	2.5	DMEM
7	(c)	2.5	DMEM
8	(d)	2.5	DMEM
9	(a)	2.5	RPMI
10	(b)	2.5	RPMI
11	(c)	2.5	RPMI
12	(d)	2.5	RPMI
13	(a)	2.5	Human plasma
14	(b)	2.5	Human plasma
15	(c)	2.5	Human plasma
16	(d)	2.5	Human plasma
17	(a)	2.5	PMNs in human plasma
18	(b)	2.5	PMNs in human plasma
19	(c)	2.5	PMNs in human plasma
20	(d)	2.5	PMNs in human plasma

**Table 6 biomolecules-14-01242-t006:** Conditions for pressure treatment.

Condition	(a) Normoxia	(b) Hyperoxia	(c) Hyberbar Oxygen (HBO)	(d) Cold Atmospheric Plasma (CAP)
Pressure	1 ATA	1 ATA	3 ATA	1 ATA
O_2_ concentration	21%	100%	100%	21% (+5 min CAP, 4 kHz)

## Data Availability

The data presented in this work are available on request from the corresponding author.

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
