# Peer review of "Effects of Pressure, Hypoxia, and Hyperoxia on Neutrophil Granulocytes"

_biomolecules, 2024, doi:10.3390/biom14101242_

Round 1

Reviewer 1 Report

Comments and Suggestions for Authors

The ms by Kraus and coworkers deals with the effect of different oxygen concentration and pressure on several parameters related to neutrophils activity.

It is evidently an interesting, but not completely original, study. Its major merit is in the accurate and comprehensive description of a significant number of biochemical and functional indexes of neutrophil granulocytes activation submitted to either hypoxia or hyperoxia both in normobaric or hyperbaric conditions. Taken individually, many of the observations reported in this paper are already known, but it could be considered worth of interest having all them reported together providing a better and more detailed picture.

In spite of these merits, this reviewer has a number of important concerns.

The major weakness, in the reviewer’s opinion is in that this paper remains within a “descriptive” frame, as it does not attempt any kind of biochemical insight of the movements of the parameters considered (no mention about transcription factors activation or gene expression). Similarly, no tentative conclusions are made about the consequences of such movements.  In other words, authors provide a number of observations but they do not tell us “what does it mean”.

1)               When dealing with isolated cultured cells, it should be considered that, from the environmental level of 20%, tissue concentration drops to about 3–4% (or less). Starting from atmospheric air to individual cells, pO2 decreases from approximatively 150 mm Hg in the upper airway to about 30 mm Hg in most tissues and finally to as low as 5 mm Hg in peripheral tissues, where these values are quite well maintained [IUBMB Life, 50, 279–289].
Therefore, maintaining cultures cells at 20% oxygen and 150 mm Hg is to be considered an hyperoxic/hyperbaric treatment rather than a normoxia. When dealing with in vitro cultured cells and oxygen fluctuations, the experimental choice should be directed toward the utilization of oxygen controlled culture hoods rather than the usual approach under atmospheric air. It would be indeed an expensive but unavoidable choice. Either, authors should have condidered (healthy) living subjects undergoing treatments in clinically controlled conditions

2)               Authors selected very specific oxygen concentrations in their experimental protocol, namely 4% for hypoxia and 83% for hyperoxia. Similarly, an atmospheric pressure equal to 3 bar (please convert in ATA) was considered hyperbaric. Authors should explain the reasons that have driven toward these choices. Do these values correspond to any specific situation or these values were set randomly and arbitrarily? If this is the case, please describe.

3)               One of the most interesting features of cell response when addressing the effects of oxygen fluctuation is indeed associated to “pulsed” exposure rather than to a single high or low concentration/pressure event. In other words, a living subjects (considering the survival to the treatment) usually returns to previous oxygen concentration/pressure values. Several papers addressed this point frequently reported as “normobaric oxygen paradox”. Similarly, the time length of the treatment is pivotal in determining the effects at tissue, cellular and whole organism level.

4)               There are a number of minor point to address. Even though the reviewer is not English mother tongue, it can be noted that, here and there, “funny” words are utilized such as “analyzation”. Please, let the text be checked by a native english.

5)               Beside the need to justify some experimental choice, the description of methods is confused and difficult to follow. I suggest a rewriting of some parts and possibly to re-organize the whole text for a better comprehension of the rationale behind the authors’ experimental design.

6)               The measurement of oxidative stress is still a very controversial issue. Not all the methods utilized can be considered reliable to provide solid data.

In conclusion,  this referee believes that this study is interesting but still incomplete and should be reinforced and corroborated by several additional experiments before publication. My personal suggestion is to consider the activation of transcription factors (Nf-kB, NRF2, HIF at least) and the gene expression pattern that underlies the observed effects on granulocytes. Possibly, the administration of pulsed oxygen variations would significantly improve the importance of the study. Publishing it “as is” would not provide a real advancement in the knowledge, not even incremental.

Comments on the Quality of English Language

Even though the reviewer is not English mother tongue, it can be noted that, here and there, “funny” words are utilized such as “analyzation”. Please, let the text be checked by a native english.

  Beside the need to justify some experimental choice, the description of methods is confused and difficult to follow. I suggest a rewriting of some parts and possibly to re-organize the whole text for a better comprehension of the rationale behind the authors’ experimental design.

Author Response

Reviewer #1, comment 1)
When dealing with isolated cultured cells , it should be considered that, from the environmental level of 20%, tissue concentration drops to about 3–4% (or less). Starting from atmospheric air to individual cells, pO2 decreases from approximatively 150 mm Hg in the upper airway to about 30 mm Hg in most tissues and finally to as low as 5 mm Hg in peripheral tissues, where these values are quite well maintained [IUBMB Life, 50, 279–289]. Therefore, maintaining cultures cells at 20% oxygen and 150 mm Hg is to be considered an hyperoxic/hyperbaric treatment rather than a normoxia. When dealing with in vitro cultured cells and oxygen fluctuations, the experimental choice should be directed toward the utilization of oxygen controlled culture hoods rather than the usual approach under atmospheric air. It would be indeed an expensive but unavoidable choice. Either, authors should have condidered (healthy) living subjects undergoing treatments in clinically controlled conditions.

Author's Reply: In our experiments only PMNs of human origin were considered. Human PMNs cannot be kept in culture, but must be processed and analyzed immediately after blood collection due to their short lifespan outside the human body. For this reason, the term “cultured cells” seems not appropriate for us.
Indeed, you are absolutely right, the oxygen concentration varies from approximatively 150 mm Hg (at a FI 0.21) in the upper airway to about 30 mm Hg in most tissues and finally to as low as 5 mm Hg in peripheral tissues. However, PMNs circulate in the blood, so the cells are confronted every minute with a maximum of 120 mmHg pO2 (at inspired O2 fraction: 0.21) in the arteries and lowest 5 mmHg O2 in the smallest venules in the tissue. Therefore, our term “normoxic” should be quite appropriate for our 20% (at normal pressure 760 mmHg, 120 mmHg = 16%; 152 mmHg = 20%) [1]. Nevertheless, to acknowledge your objections, we added this important issue to the introduction of our manuscript.
…to be considered an hyperoxic/hyperbaric treatment rather than a normoxia…
In our live cell imaging experiments only pressure levels of 1 atm were used. The term “hyperbaric” is therefore not applicable here.
…utilization of oxygen controlled culture hoods…
Thank you for mentioning this point we now added the explicit information in the methods section (4.6 Microscopy). That is exactly what we used in our microscopic experimental setup, as already suggested shortly (in 4.7, first paragraph after Fig. 13) or just referenced to Doblinger et al.

Reviewer #1, comment 2)
Authors selected very specific oxygen concentrations in their experimental protocol, namely 4% for hypoxia and 83% for hyperoxia. Similarly, an atmospheric pressure equal to 3 bar (please convert in ATA) was considered hyperbaric. Authors should explain the reasons that have driven toward these choices. Do these values correspond to any specific situation or these values were set randomly and arbitrarily? If this is the case, please describe.
Author's Reply: We selected specific oxygen concentrations of 4% for hypoxia and 83% for hyperoxia based on practical limitations of the experimental setup. The 4% concentration was the lowest achievable level in the microscope climate chamber, ensuring that the conditions accurately reflected a hypoxic environment. On the other hand, the atmospheric pressure of 3 ATA was the maximum pressure attainable in the small pressure chamber used for the experiments, which allowed for the simulation of hyperbaric conditions. These values were not chosen arbitrarily; rather, they were determined by the technical constraints of the equipment and the need to create controlled experimental conditions that could effectively mimic specific physiological states (see response to comment #1).
…please convert in ATA…
The term “bar” has been replaced by “ATA” throughout the manuscript, including all tables and figures.

Reviewer #1, comment 3)
One of the most interesting features of cell response when addressing the effects of oxygen fluctuation is indeed associated to “pulsed” exposure rather than to a single high or low concentration/pressure event. In other words, a living subjects (considering the survival to the treatment) usually returns to previous oxygen concentration/pressure values. Several papers addressed this point frequently reported as “normobaric oxygen paradox”. Similarly, the time length of the treatment is pivotal in determining the effects at tissue, cellular and whole organism level.
Author's Reply: We agree, the aspect of the normobaric oxygen paradox is an issue that we unfortunately have not paid any attention to so far, but which is well worth to be addressed in our work. For this reason, we have included a paragraph on this in our discussion.

Reviewer #1, comment 4)
There are a number of minor point to address. Even though the reviewer is not English mother tongue, it can be noted that, here and there, “funny” words are utilized such as “analyzation”. Please, let the text be checked by a native english.
Author's Reply: The text was read by an English native speaker and any linguistic errors were corrected.

Reviewer #1, comment 5)
Beside the need to justify some experimental choice, the description of methods is confused and difficult to follow. I suggest a rewriting of some parts and possibly to re-organize the whole text for a better comprehension of the rationale behind the authors’ experimental design.
Author's Reply: Important parts of the manuscript have been rewritten and thus reorganized. To address your main point of criticism, we have defined the experimental groups again at the end of the introduction and at the beginning of the material and methods section (section 4). In addition we inserted the experimental workflow graphic - as suggested by your co-reviewer- at the beginning of section 4.

In order to present the results in a clearer and more comprehensible way, we added our intention (why we chose the respective experiments and what we wanted to measure with them) at the beginning of each results section. We hope this approach, which was requested by your co-reviewer- has made the text much easier to understand.

We agree, that the results and methods section in particular presented a considerable challenge in terms of readability. To enhance the contextualization of the results, we listed the reason for the measurements at the beginning of each section in the results chapter. Moreover, we summarize the key observations at the end of each subsection. We hope, that this approach will not only highlight the most important findings but also will help maintain the flow of ideas and reinforce the research direction. By emphasizing the critical aspects of the data set, the results should be made easier to understand for the reader.

Moreover, we have checked the descriptions of figures and tables and supplemented them to make them easier to understand. We hope that this also contributed significantly to an improved comprehensibility.

We hope that we have been able to clear up any confusion in this way. If there are still unclear passages, we would be pleased if you could indicate these precisely (with line numbers). We can then address these sections specifically.

Reviewer #1, comment 6)
The measurement of oxidative stress is still a very controversial issue. Not all the methods utilized can be considered reliable to provide solid data.

Author's Reply: We agree that measurement of oxidative stress is difficult and different approaches exist. Numerous well-established cellular and biochemical assays—among others those reviewed by Toetsch et al.—provide great insight into migration and biochemical processes that enable neutrophil immune responses [2]. However, many of these assays on in vitro chemotaxis and migration investigated cells in environments that considerably differed from those in vivo.

To simulate conditions that cells find on vessel walls, various assays (including assays by Dunn, Zigmond, Boyden, Gundersen and Barrett) work with a two-dimensional environment in which mobile cells come into contact with glass or filters [3, 4].

Nevertheless, PMN are generally distributed in three-dimensional human tissue in vivo; according to Parkhurst et al., it is therefore more appropriate to investigate extracellular influence on PMN in a more realistic environment such as a three-dimensional experimental set-up to reproduce the conditions of the human body in the best possible manner [3, 4]. Since three-dimensional chemotaxis experiments attempt to reconstruct environments of the human body as accurately as possible, we consciously conducted research on PMN function in different three-dimensional matrices to better understand neutrophil behavior in extracellular environments [5, 6]. The IBIDI μSlide chambers used in this work were suitable for the simulation of in vivo conditions of the interstitium because they allowed the cells to be embedded in a matrix [3, 7].

Apart from steric aspects of the in vitro model, temporal aspects of the examination methods have also to be taken into consideration. We view the continuous recording of immune effects during the entire experimental observation period as a great advantage of our method. Many reliable assays have measured neutrophil ROS generation and MPO and NET release. But just as in the case of flow cytometry, as used by Ramirez et al. for instance, the methods used in these assays were mainly snap-reading that—unlike our method—lack uninterrupted progressions [8, 9].

Furthermore, our experimental set-up involving live-cell imaging allowed the investigator to diversify direct stimuli as well as directional chemo-attraction in any desired manner. The possibility of objectively analyzing these processes in a single set-up is a great advantage in contrast to earlier studies in which solely flow cytometry was used to analyze PMN behavior [9, 10]. In addition, simultaneous observation enables a comparison of each activity in relation to each other. Apart from minor modifications, Doblinger et al., Bredthauer et al., Pai et al., Kolle et al. and recently Hundhammer et al. used the same method as described in our manuscript for simultaneous observation of ROS production, MPO release and NETosis. Thus, it is proven that this method is sufficient to observe the three neutrophil immune effects [8, 10–13].

Despite the limitations described in section 3.5, it should be noted that the methods used in our experiments were appropriate for the functional testing of neutrophils in different extracellular matrices outside the human body.

Reviewer #1, comment 7)
In conclusion, this referee believes that this study is interesting but still incomplete and should be reinforced and corroborated by several additional experiments before publication. My personal suggestion is to consider the activation of transcription factors (Nf-kB, NRF2, HIF at least) and the gene expression pattern that underlies the observed effects on granulocytes. Possibly, the administration of pulsed oxygen variations would significantly improve the importance of the study. Publishing it “as is” would not provide a real advancement in the knowledge, not even incremental.

PMNs, as short-lived, fully differentiated, post-mitotic cells, generally show little protein synthesis capability and rely almost entirely on existing stores for their activities. We are convinced, that the quantification of transcription factors will yield little benefit for the understanding of oxygen impact on PMNs:
Mature PMNs emerge in the blood devoid of any proliferative capacity, but fully capable of launching an immune response. The difference in protein contents among neutrophil granule subsets is not driven by protein sorting. Instead, different granules are sequentially formed during neutrophil differentiation in what has been described as a targeting by timing model. According to this model, as different granule proteins are synthesized during different stages of neutrophil differentiation several granule subsets are generated. The rapid immunological response of neutrophils does not depend particularly on pre-formed granule proteins with antimicrobial activity but rather on the ability to generate respiratory burst activity [14].

Neutrophils, upon stimulation undergo a series of immediate changes without the need for de novo synthesis of proteins. Exocytosis, also known as degranulation in neutrophil, is the release pre-formed and stored mediators from granules [14].

As we have already stated in the discussion of our manuscript, HIF-1α is a global regulator of the cellular response preferential to low oxygen and NET formation showed no changes dependent to the oxygen conditions in our experiments, we suggest that HIF-1α is not solely responsible for the observed reactions. Nevertheless, as already specified by Branitzki et al. our experimental setup with changing oxygen conditions between hypo- and hyperoxia might not be the ideal tool to analyze HIF-1α.
In alignment with this, new experiments of our laboratory (data unfortunately unpublished to date) demonstrated that even when exposing PMNs to PMA or centrifugal shear forces, there is minimal to no metabolic change observed. Consequently, it is not foreseeable that a slight change in the oxygen content will lead to significant changes of transcription factors. Therefore, we remain of the conviction that considering the activation of transcription factors will not provide any leading-edge insights or sufficient explanation for the observed PMN behavior and therefore will not provide any significant benefit to the manuscript.
Nevertheless, we would like to take this opportunity to thank you for your constructive suggestions for improvement and hope that our changes meet your expectations.

References
[1] Heil W, Koberstein R, Zawta B. Referenzbereiche für Kinder und Erwachsene: Präanalytik. Mannheim: Roche Diagnostics GmbH; 2004.
[2] Toetsch S, Olwell P, Prina-Mello A, Volkov Y. The evolution of chemotaxis assays from static models to physiologically relevant platforms. Integr Biol (Camb) 2009; 1(2): S. 170–181.
[3] Parkhurst MR, Saltzman WM. Quantification of human neutrophil motility in three-dimensional collagen gels. Effect of collagen concentration. Biophysical Journal 1992; 61(2): S. 306–315.
[4] Zengel P, Nguyen-Hoang A, Schildhammer C, Zantl R, Kahl V, Horn E. mu-Slide Chemotaxis: A new chamber for long-term chemotaxis studies. BMC Cell Biol 2011; 12: S. 21.
[5] Keenan TM, Folch A. Biomolecular gradients in cell culture systems. Lab Chip 2007; 8(1): S. 34–57.
[6] Islam LN, McKay IC, Wilkinson PC. The use of collagen or fibrin gels for the assay of human neutrophil chemotaxis. J Immunol Methods 1985; 85(1): S. 137–151.
[7] Kim BJ, Wu M. Microfluidics for Mammalian Cell Chemotaxis. Annals of Biomedical Engineering 2012; 40(6): S. 1316–1327.
[8] Doblinger N, Bredthauer A, Mohrez M, Hähnel V, Graf B, Gruber M et al. Impact of hydroxyethyl starch and modified fluid gelatin on granulocyte phenotype and function. Transfusion 2019; 59(6): S. 2121–2130.
[9] Ramirez GA, Manfredi AA, Rovere-Querini P, Maugeri N. Bet on NETs! Or on How to Translate Basic Science into Clinical Practice. Front. Immunol. 2016; 7: S. 417.
[10] Pai D, Gruber M, Pfaehler S-M, Bredthauer A, Lehle K, Trabold B. Polymorphonuclear Cell Chemotaxis and Suicidal NETosis: Simultaneous Observation Using fMLP, PMA, H7, and Live Cell Imaging. Journal of Immunology Research 2020: S. 1–10.
[11] Kolle G, Metterlein T, Gruber M, Seyfried T, Petermichl W, Pfaehler S-M et al. Potential Impact of Local Anesthetics Inducing Granulocyte Arrest and Altering Immune Functions on Perioperative Outcome. J Inflamm Res 2021; 14: S. 1–12.
[12] Bredthauer A, Kopfmueller M, Gruber M, Pfaehler S-M, Lehle K, Petermichl W et al. Therapeutic Anticoagulation with Argatroban and Heparins Reduces Granulocyte Migration: Possible Impact on ECLS-Therapy? Cardiovascular Therapeutics 2020; (26): S. 1–10.
[13] Hundhammer T, Gruber M, Wittmann S. Paralytic Impact of Centrifugation on Human Neutrophils. Biomedicines 2022; 10(11).
[14] Sheshachalam A, Srivastava N, Mitchell T, Lacy P, Eitzen G. Granule protein processing and regulated secretion in neutrophils. Front Immunol 2014; 5: S. 448.

You can also see the reply in the attachment.

Reviewer 2 Report

Comments and Suggestions for Authors

Dear Authors.

The manuscript submitted by you, whose title is 'Effects of pressure, hypoxia, and hyperoxia on neutrophil granulocytes' is of great interest to the area of research you are working on, being an excellent experimental design, and a logical sequence of questions and experiments that gives fluidity to work submitted by you for review. I will send you my comments so you can improve your manuscript's strength and quality.

Overall, this is an excellent manuscript. It is well-written in most sections and has outstanding development, mainly in the introduction and discussion sections. Still, more work needs to be done in the methodology sections, and you should work on the results section. Singular citation style in some cases, please clarify why after 'some citations' there is a p and a number in brackets (i.e., p 679, line 76), which I assume is the page of interest in the cited publication.

The introduction section is fluid and easy to read, and it gets to the general research question excellently, making it clear what the logical flow of ideas is that gives rise to the research question. 

The introductory section is smooth and agile to read, getting from general to the research question excellently, making clear the logical flow of ideas that give rise to the research question. 

Regarding the methodology, the methodologies, techniques, and times descriptions are excellent and enable someone to reproduce the experiments; this section has perfect choices of figures and tables. However, please respond and introduce in the text the identification and definition of each experimental group and what experiments were applied to them (suggestion: the first figure of the manuscript should be the first figure of the methodology section). In addition, the intentions of each of the methodologies are more useful at the beginning of the results section of each of them or their methodology section; the authors usually added them in the discussion section. The statistical analysis is consistent, and the mention to ethics committee's approval was appropiate. The sample size of each experimental group should be clarified, as the total number of experiments is mentioned, but it is unclear how many individuals these numbers come from. This sample size would make the dispersions of the figures and tables more straightforward and more precise if this sample is adequate for statistical analysis.

The results section was the most complex to read. Although it is very descriptive, it mentions each of the results, statistically significant or not, which is excellent but not adequate due to the length and the fact that it needs to be more accurate in the objective that these results contribute to answering the research question. It needs to improve the overall look or contextualization of the result, making the results sound less meaningful and losing the flow of ideas and the research direction. A suggestion might be to summarise or address the results at the end of each section, emphasizing what you want to highlight or give importance to in that data set.

Regarding the figures and tables in this section, the figure captions could be more informative of the generated or tabulated data. A self-explanatory figure has not been generated, and the descriptions in the text do not do so either, so the importance of showing or giving them the importance of why they should be a figure or table is lost. Please correct the figures and figure captions and describe them better. It will give the significance of your results, and the reader could more easily conclude your analysis.

The description of the discussion section was excellent. I have no comments or corrections on this section.

The manuscript could acquire greater soundness and quality by including comments and suggestions.

I approve with major revisions its publication.

Author Response

Reviewer #2, comment 1)
Still, more work needs to be done in the methodology sections, and you should work on the results
section. Singular citation style in some cases, please clarify why after 'some citations' there is a p and
a number in brackets (i.e., p 679, line 76), which I assume is the page of interest in the cited publication.
Author's Reply: We have solved the problem of inconsistent references so that coherent references can now be found throughout the whole manuscript. 

Reviewer #2, comment 2)
However, please respond and introduce in the text the identification and definition of each experimental group and what experiments were applied to them (suggestion: the first figure of the manuscript should be the first figure of the methodology section). 
Author's Reply: For better identification and definition, we have defined the experimental groups again at the beginning of the material and methods section (section 4) and inserted the graphic - as you suggested - at this point of the text. 

Reviewer #2, comment 3)
In addition, the intentions of each of the methodologies are more useful at the beginning of the results section of each of them or their methodology section; the authors usually added them in the discussion section. 
Author's Reply: In order to present the results in a clearer and more comprehensible way, we added our intention (why we chose the respective experiments and what we wanted to measure with them) at the beginning of each results section. We hope that this has made the text much easier to
understand.

Reviewer #2, comment 4)
The sample size of each experimental group should be clarified, as the total number of experiments is
mentioned, but it is unclear how many individuals these numbers come from. This sample size would
make the dispersions of the figures and tables more straightforward and more precise if this sample is
adequate for statistical analysis. 
Author's Reply: Each experiment was carried out with a different individual, so that the number of experiments can be equated with the test persons. We have added supplementary information of the
participating test persons in section 4.1. 

Reviewer #2, comment 5)
The results section was the most complex to read. Although it is very descriptive, it mentions each of
the results, statistically significant or not, which is excellent but not adequate due to the length and the fact that it needs to be more accurate in the objective that these results contribute to answering the research question. It needs to improve the overall look or contextualization of the result, making the results sound less meaningful and losing the flow of ideas and the research direction. A suggestion might be to summarise or address the results at the end of each section, emphasizing what you want to highlight or give importance to in that data set. 

Author's Reply: We agree, that the results section presented a considerable challenge in terms of readability. To enhance the contextualization of the results, we listed the reason for the measurements at the beginning of each section in the results chapter. Moreover, we summarize the key
observations at the end of each subsection. We hope, that this approach will not only highlight
the most important findings but also -as suggested by you- will help maintain the flow of ideas
and reinforce the research direction. By emphasizing the critical aspects of the data set, the
results should be made easier to understand for the reader.

Reviewer #2, comment 6): Regarding the figures and tables in this section, the figure captions could be more informative of the generated or tabulated data. A self-explanatory figure has not been generated, and the descriptions in the text do not do so either, so the importance of showing or giving them the importance of why they should be a figure or table is lost. Please correct the figures and figure captions and describe them better. It will give the significance of your results, and the reader could more easily conclude your analysis. 

Author's Reply: We have checked the descriptions of the figures and tables and supplemented them to make them easier to understand. We hope that this has significantly improved comprehensibility.
We would like to take this opportunity to thank you for your comments and suggestions for
improvement. We are convinced that the changes you have recommended will significantly
improve the readability of the manuscript. We hope, that we have been able to address your
expectations.

You can also find the reply in the attachment.

Round 2

Reviewer 1 Report

Comments and Suggestions for Authors

I read carefully the rebuttal. Authors replied to all my comments but they did not consider any of them. Their justifications are actually very vague and not scientifically very sound.

References related to the "return to normoxia" after hyper of hypoxia are missing. Similarly, literature related to the importance of transcription factors activation and, downstream, gene expression is missing.
